# Distinct spatial immune microlandscapes are independently associated with outcomes in triple-negative breast cancer

Jodi M. Carter[1,17], Saranya Chumsri[2,17], Douglas A. Hinerfeld[3,17], Yaohua Ma[4], Xue Wang [4], David Zahrieh[5], David W. Hillman [5], Kathleen S. Tenner[5], Jennifer M. Kachergus[6], Heather Ann Brauer[7], Sarah E. Warren[8], David Henderson[9], Ji Shi[6], Yi Liu[6], Heikki Joensuu [10], Henrik Lindman[11], Roberto A. Leon-Ferre[12], Judy C. Boughey [13], Minetta C. Liu [14], James N. Ingle [12], Krishna R. Kalari[5], Fergus J. Couch[15], Keith L. Knutson [16], Matthew P. Goetz[12,18], Edith A. Perez[2,18] & E. Aubrey Thompson [6,18] ✉

The utility of spatial immunobiomarker quantitation in prognostication and therapeutic prediction is actively being investigated in triple-negative breast cancer (TNBC). Here, with high-plex quantitative digital spatial profiling, we map and quantitate intraepithelial and adjacent stromal tumor immune protein microenvironments in systemic treatment-naïve (female only) TNBC to assess the spatial context in immunobiomarker-based prediction of outcome. Immune protein profiles of CD45-rich and CD68-rich stromal microenvironments differ significantly. While they typically mirror adjacent, intraepithelial microenvironments, this is not uniformly true. In two TNBC cohorts, intraepithelial CD40 or HLA-DR enrichment associates with better outcomes, independently of stromal immune protein profiles or stromal TILs and other established prognostic variables. In contrast, intraepithelial or stromal microenvironment enrichment with IDO1 associates with improved survival irrespective of its spatial location. Antigen-presenting and T-cell activation states are inferred from eigenprotein scores. Such scores within the intraepithelial compartment interact with PD-L1 and IDO1 in ways that suggest prognostic and/or therapeutic potential. This characterization of the intrinsic spatial immunobiology of treatment-naïve TNBC highlights the importance of spatial microenvironments for biomarker quantitation to resolve intrinsic prognostic and predictive immune features and ultimately inform therapeutic strategies for clinically actionable immune biomarkers.

In the past decade, growing evidence has established that intrinsic host-tumor immune responses can dictate clinical outcomes and predict therapeutic responses in patients with triple-negative breast cancer (TNBC). Several studies have demonstrated that expression of programmed death-ligand 1 (PD-L1)[1], was associated with improved outcomes in TNBC. Furthermore, PD-L1 positivity was also associated with better responses to immune checkpoint inhibitors in patients with metastatic TNBC treated with pembrolizumab[2] and atezolizumab[3]. However, the clinical benefit of immune checkpoint inhibitors has been modest in patients with metastatic TNBC, and atezolizumab was

voluntarily withdrawn from the U.S. market due to limited benefit. More recently, pembrolizumab was also approved in combination with anthracycline-based chemotherapy in the neoadjuvant setting for patients with high-risk early-stage TNBC[4]. However, tumoral PD-L1 status was not predictive of pembrolizumab responsiveness in the early-stage setting[5], highlighting the need for optimized tissue-based biomarkers to inform patient selection for immunotherapies[2, 3].

Beyond the PD-(L)1 axis, stromal tumor-infiltrating lymphocytes (sTILs) also have established prognostic significance in TNBC. Recent data have demonstrated that high sTILs are associated with improved survival and may identify patients eligible for immunotherapeutic intervention[6–11]. However, stromal TIL scoring does not provide insight into the composition or biological function of stromal and intraepithelial immune cell repertoires. Similarly, computational deconvolution of bulk RNA analysis or single cell gene expression analyses are surrogates of protein expression and lack spatial context[12, 13]. The importance of spatial context in tumoral immune microenvironments (TIME) is underscored by recent data describing distinct immune phenotypes in TNBC[14–16], based on both the location and quantity of immune cell infiltration: inflamed, immune-excluded, and immune-desert tumors[17]. In contrast to other phenotypes, the immune-excluded phenotype retains immune cells in stroma without intraepithelial tumoral penetration, and this relationship may underlie the lack of clinically meaningful responses to PD-(L)1checkpoint inhibitors[3, 17–21]. In any event, these phenotypes require a detailed and comprehensive immune microenvironment characterization, including quantitation of cell types and their locations, target protein/biomarker target quantitation, and location of immune cells to provide biological insight and inform therapeutic strategies. The development of such strategies requires a more detailed understanding of the numbers, types, activities, and locations of immune cells within the tumor microenvironment.

The GeoMx™ Digital Spatial Profiling (DSP) platform has a high multiplexing capacity, with simultaneous digital quantification of multiple UV-photocleavable oligonucleotide-conjugated antibodies bound to regions of interest (ROIs), with a wide dynamic range and high reproducibility[22–26]. We have previously used this platform to characterize PD-L1-associated immune protein profiles in TNBC[24, 27]. Here, using the GeoMx™ platform, we map and quantify dozens of immune proteins in two cohorts of systemic therapy-naïve TNBC to (1) characterize intraepithelial and adjacent stromal TIME, (2) delineate the impact of different immune-rich stroma on intraepithelial TIME, and (3) assess the spatial context in immunobiomarker-based prediction of clinical outcomes in TNBC. Our data demonstrate that spatial context influences immune protein microenvironments, with substantive differences between targeted intraepithelial segments and adjacent stromal protein TIME. Intraepithelial antigen presentation-related immune proteins (HLA-DR and CD40) are independently prognostic of clinical outcomes, and functional antigen presentation and T-cell activation eigenprotein scores inform the prognostic context of immune checkpoint biomarkers (e.g., PD-L1 and IDO1). These data provide insight into existing and emerging immune-based therapies and their companion diagnostic assays, and complement existing immune-based prognosticators in TNBC, such as sTILs, to optimize patient selection for targeted therapeutics.

## Results

### Intraepithelial and stromal immune protein (micro)landscapes in early-stage TNBC

Full-face tumor sections from 44 TNBC (FinXX trial) were matched for tumor size, lymph node status, treatment arm, and patient age between a subset that recurred ($N = 22$) or did not recur ($N = 22$). Median follow-up time was 2.3 years (range 0.2–8.3 years) in the patients that recurred (recurrence-free survival (RFS) = NO) and 10.7 years (range 9.0–11.8 years) for patients with durable RFS

(RFS = YES) (Supplementary Table 1). To begin, we quantified immune proteins in the spatially defined intraepithelial tumor and (adjacent) stromal microenvironments. Segment-level normalized counts for each region of interest (ROI) in each sample (Supplementary Data 1) and tumor-average normalized counts (average of four ROIs/tumor) for 29 proteins detected above background in either intraepithelial or stroma segment averages (Supplementary Data 2) are provided. Predictably, intraepithelial tumor segments had significantly more cytokeratin and Ki-67 than stroma segments (log2FC: 3.0 and 1.0, respectively, Supplementary Fig. 1), and these two proteins were among the most abundant proteins in intraepithelial segments (Supplementary Data 3). Conversely, all immune proteins, except CD56 and CD127, were enriched in stroma compared to intraepithelial segments, and the most abundant proteins overall in the stroma were SMA, CD45, fibronectin, CD44, and HLA-DR (Supplementary Data 3).

### Stromal CD45+ or CD68+ immune cell enrichment impacts microenvironments

We next evaluated how stromal immune cell enrichment (CD45EnR or CD68EnR) impacted the immune protein profiles of stroma or intraepithelial tumor segments. Stroma segments from CD68-enriched ROIs and CD45-enriched ROIs differed substantially in their immune protein abundance (Supplementary Fig. 2). Predictably, CD45EnR Stroma was enriched in B and T-cell markers, as well as IDO1, with CD20 showing the highest differential expression [log 2FC: 2.1 (range: 1.7–2.5)], followed by CD3 and CD8. As expected, CD68EnR Stroma had more CD68 protein than CD45EnR Stroma [mean log2FC: 1.9 (range 1.6–2.3)]. CD68EnR Stroma segments also had higher levels of immune checkpoint-related proteins CTLA4, B7-H3, and TIM-3, and extracellular matrix/(myo)fibroblast activation proteins (fibronectin and SMA) compared to CD45EnR Stroma. PD-L1 and PD-L2 were similarly abundant in CD45EnR Stroma and CD68EnR Stroma (and were low-abundance proteins, Supplementary Data 3).

We next evaluated the relationship between the stromal immune protein profile and adjacent intraepithelial tumor segments. CD45EnR Intraepithelial segments mirrored their adjacent stromal counterparts, enriched in many lymphocytic markers compared to CD68EnR Intraepithelial segments, including CD3, CD4, CD8, CD11c, CD20, CD44, CD45, STING, B7-H3, and TIM-3 (Supplementary Fig. 3). However, CD68 protein in intraepithelial segments adjacent to CD68EnR Stroma was not significantly more abundant than that observed in intraepithelial segments adjacent to CD45EnR Stroma [log2 FC = 0.3 (−0.2 to 0.8) $p = 0.21$]. This observation suggests that the extent to which CD68-expressing macrophages infiltrate the intraepithelial compartment is not strictly tied to the abundance of such cells in the tumor-adjacent stroma. Predictably, both CD45EnR and CD68EnR intraepithelial segments had higher levels of most immune proteins compared to TumorEnR intraepithelial segments (which lacked immune-dense adjacent stroma) (Supplementary Figs. 4 and 5), suggesting that the stromal immune cell density does impact the immune landscape of the adjacent intraepithelial tumor nests. One notable exception was CD56, which was more or less equally abundant in all intraepithelial segments, regardless of the immune cell composition of the adjacent stroma.

### Spatial context determines immune protein association with outcome

Our initial analysis focused on the identification of proteins that were differentially expressed in a spatially defined manner in tumors that recurred or did not recur. We identified 11 immune proteins that were significantly more abundant in tumors from patients with durable RFS (RFS = YES) vs. patients that recurred (RFS = NO) in one or more intraepithelial segment classes [fold change, after log2 transformation (log2 FC) = RFS YES/NO] after multiple testing correction [Benjamini–Yekutieli adjusted (BYadj) $p < 0.05$, italicized, Table 1]. Only one protein

 

**Table 1 | Differential expression of immune proteins as a function of spatial distribution in intraepithelial and stromal segments and recurrence**

| Protein | Intraepithelial (cytokeratin-positive) segments | | | | | | | | Stroma (cytokeratin-negative) segments | | | | | |
|---|---|---|---|---|---|---|---|---|---|---|---|---|---|---|
| | All segments fold change | p-value | CD45ENR fold change | p-value | CD68ENR fold change | p-value | TumorENR fold change | p-value | All segments fold change | p-value | CD45ENR fold change | p-value | CD68ENR fold change | p-value |
| B7-H3 | 1.2 | 0.038 | 1.2 | 0.212 | 1.2 | 0.237 | 1.2 | 0.249 | 1.0 | 0.784 | 0.9 | 0.234 | 1.0 | 0.841 |
| B2M | **1.7** | 0.000 | **1.8** | 0.000 | **1.6** | 0.000 | **1.6** | 0.000 | 1.3 | 0.000 | 1.2 | 0.013 | 1.3 | 0.001 |
| CD1c | 1.4 | 0.000 | **1.8** | 0.000 | 1.2 | 0.116 | 1.0 | 0.630 | 1.0 | 0.519 | 1.2 | 0.126 | 0.9 | 0.493 |
| CD127 | 1.0 | 0.373 | 1.0 | 0.700 | 0.9 | 0.380 | 1.0 | 0.745 | 1.2 | 0.013 | 1.0 | 0.613 | 1.0 | 0.992 |
| CD20 | **2.1** | 0.000 | **2.5** | 0.000 | **1.6** | 0.000 | **1.7** | 0.000 | 1.3 | 0.018 | 1.1 | 0.559 | 1.2 | 0.128 |
| CD25 | 1.3 | 0.000 | 1.4 | 0.001 | 1.2 | 0.022 | 1.3 | 0.011 | 1.2 | 0.000 | 1.1 | 0.181 | 1.1 | 0.109 |
| CD3 | **1.6** | 0.000 | **1.8** | 0.000 | 1.3 | 0.063 | 1.3 | 0.116 | 1.1 | 0.216 | 1.1 | 0.196 | 1.0 | 0.783 |
| CD4 | 1.3 | 0.000 | 1.5 | 0.000 | 1.2 | 0.145 | 1.1 | 0.364 | 1.0 | 0.629 | 1.0 | 0.699 | 0.9 | 0.434 |
| CD40 | **1.9** | 0.000 | **1.9** | 0.000 | **1.8** | 0.000 | **2.2** | 0.000 | 1.2 | 0.004 | 1.1 | 0.142 | 1.4 | 0.003 |
| CD44 | 1.3 | 0.024 | 1.1 | 0.653 | 1.3 | 0.183 | 1.5 | 0.029 | 1.0 | 0.723 | 0.9 | 0.507 | 1.2 | 0.189 |
| CD45 | 1.4 | 0.001 | 1.5 | 0.000 | 1.1 | 0.341 | 1.2 | 0.325 | 1.1 | 0.413 | 1.1 | 0.489 | 1.1 | 0.368 |
| CD56 | **3.1** | 0.000 | **2.9** | 0.000 | **2.5** | 0.000 | **4.1** | 0.000 | 1.3 | 0.000 | 1.2 | 0.002 | 1.3 | 0.003 |
| CD68 | 1.0 | 0.946 | 1.1 | 0.300 | 0.9 | 0.619 | 0.9 | 0.647 | 0.9 | 0.080 | 0.8 | 0.036 | 0.9 | 0.275 |
| CD8 | 1.5 | 0.000 | **1.7** | 0.000 | 1.2 | 0.175 | 1.5 | 0.004 | 1.1 | 0.259 | 1.1 | 0.404 | 1.1 | 0.654 |
| CTLA4 | 1.2 | 0.000 | 1.4 | 0.001 | 1.0 | 0.744 | 1.3 | 0.001 | 0.8 | 0.013 | 0.9 | 0.412 | 0.6 | 0.000 |
| FN | 1.4 | 0.005 | 1.4 | 0.083 | **1.9** | 0.001 | 0.9 | 0.454 | **1.6** | 0.000 | 1.1 | 0.524 | **2.1** | 0.000 |
| GZMB | **1.9** | 0.000 | **1.9** | 0.000 | **1.8** | 0.000 | **1.8** | 0.000 | **1.6** | 0.000 | 1.5 | 0.000 | 1.5 | 0.000 |
| HLA-DR | **2.8** | 0.000 | **2.4** | 0.000 | **2.6** | 0.000 | **3.7** | 0.000 | 1.0 | 0.556 | 1.0 | 0.766 | 1.0 | 0.916 |
| ICOS | **1.9** | 0.000 | **2.0** | 0.000 | **1.8** | 0.000 | **1.7** | 0.000 | 1.2 | 0.004 | 1.1 | 0.242 | 1.4 | 0.003 |
| IDO1 | **1.8** | 0.000 | **2.2** | 0.000 | **1.6** | 0.034 | **1.6** | 0.029 | **1.8** | 0.000 | **1.5** | 0.004 | 2.7 | 0.000 |
| Ki-67 | 1.2 | 0.052 | 1.2 | 0.224 | 1.3 | 0.050 | 1.0 | 0.930 | 1.1 | 0.351 | 1.1 | 0.473 | 1.3 | 0.030 |
| PanCk | 1.0 | 0.639 | 1.1 | 0.533 | 0.9 | 0.617 | 1.1 | 0.510 | 1.2 | 0.041 | 1.1 | 0.331 | 1.0 | 0.670 |
| PD-L1 | **1.7** | 0.000 | **1.9** | 0.000 | **1.6** | 0.000 | **1.6** | 0.000 | 1.5 | 0.000 | 1.3 | 0.001 | 1.5 | 0.000 |
| PD-L2 | **4.0** | 0.000 | **3.9** | 0.000 | **3.6** | 0.000 | **4.4** | 0.000 | **3.3** | 0.000 | **3.2** | 0.000 | 3.2 | 0.000 |
| SMA | 1.3 | 0.006 | 1.5 | 0.011 | 0.9 | 0.369 | **1.8** | 0.001 | 0.7 | 0.000 | 0.8 | 0.058 | **0.5** | 0.000 |
| STING | 1.2 | 0.012 | 1.4 | 0.000 | 0.9 | 0.602 | 1.1 | 0.326 | 1.0 | 0.940 | 1.0 | 0.639 | 1.1 | 0.235 |
| TGFB1 | **1.6** | 0.000 | **2.0** | 0.000 | 1.5 | 0.000 | 1.4 | 0.000 | 1.2 | 0.001 | 1.2 | 0.014 | 1.1 | 0.181 |
| Tim-3 | 1.0 | 0.591 | 1.2 | 0.109 | 1.0 | 0.692 | 1.0 | 0.686 | 0.9 | 0.289 | 0.9 | 0.390 | 0.9 | 0.417 |
| VISTA | 1.4 | 0.000 | 1.5 | 0.001 | 1.3 | 0.027 | 1.3 | 0.042 | 1.3 | 0.000 | 1.0 | 0.911 | 1.2 | 0.019 |

Differential expression, given as estimated fold change (FC), was calculated from normalized protein counts for RFS = YES/RFS = NO tumors. The linear mixed model was used to calculate the estimated fold change and significance, defined as nominal two-sided $p < 0.05$ with FC > 1.5 or < 0.5, shown in bold font. Proteins with Benjamini–Yekutieli adjusted $p < 0.05$ in one or more segment types are shown in italic font. PD-L2 alone was significant at BYadj $p < 0.05$ in any stroma segment (AllSegments, stroma and CD45EnR, stroma). "All Intraepithelial" data are averaged for all categories of intraepithelial segments (CD45EnR, CD68EnR, and TumorEnr, $n = 12$ ROI/sample). "All Stroma" are averaged data from stromal segments (CD45EnR and CD68EnR, $n = 8$ ROI/sample). Source data are given in Supplementary Data 1.
B2M beta-2-microglobulin, FN fibronectin, GZMB granzyme B, PanCK pan-cytokeratin, SMA smooth muscle actin.

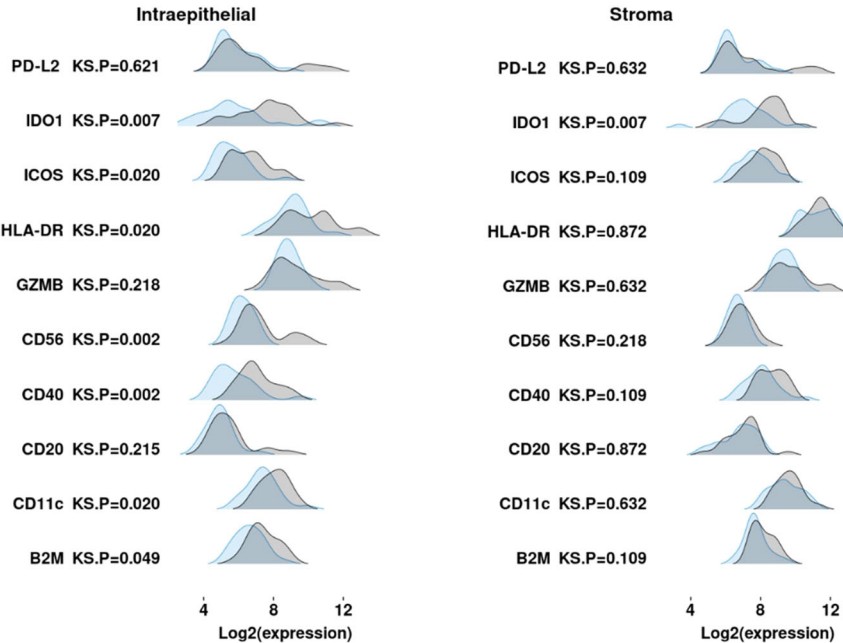

**Fig. 1 | Distribution of key immune proteins in intraepithelial and stromal segments as a function of recurrence.** Distribution of tumor-average log2 transformed normalized counts from 10 immune proteins (differentially expressed at BYadj $p < 0.05$) for both intraepithelial and stromal segments from FinXX samples. Tumors that did not recur (RFS = YES, $n = 22$) are shown in light gray, whereas tumors that did recur (RFS = NO, $n = 22$) are shown in blue. Two-sided $p$-values were calculated using the Kolmogorov–Smirnov (KS) model to test the null hypothesis that the distribution of protein expression in the two groups of samples was the same. GZMB granzyme B, B2M beta-2 microglobulin. Source data are given in Supplementary Data 1.

(PD-L2) was overexpressed as a function of RFS at BYadj $p < 0.05$ in any stromal segment class. Intraepithelial beta-2-microglobulin (B2M), CD11c, CD20, CD40, CD56, granzyme B (GZMB), HLA-DR, ICOS, IDO1, and PD-L2 were overexpressed in RFS = YES samples but ranged in absolute abundance from high-abundance (HLA-DR) to low-abundance (CD56) proteins (Supplementary Data 3). Intraepithelial B2M, CD40, CD56, GZMB, HLA-DR, ICOS, IDO1, PD-L1, and PD-L2 were more abundant in tumors that did not recur across all three classes of intraepithelial segments (CD45EnR and CD68EnR and TumorEnR, Table 1). Although TGFβ1 was differentially expressed, this antibody was deleted from subsequent reagent releases because of concerns about specificity.

The distribution of expression of 10 differentially expressed proteins in intraepithelial segments from tumors that did not recur (RFS = YES in Fig. 1) suggests that the spatial context and quantity of these proteins impact the immune surveillance and effector function of these key immune proteins (antigen presentation, PD-(L)1 and other immune checkpoint proteins, and GZMB).

Kaplan–Meier (KM) analysis was carried out to assess the outcome in tumors with high, mid, and low tertile abundance of HLA-DR or IDO1 (Fig. 2). HLA-DR was not significantly associated with the outcome when all segments (intraepithelial + stromal) were combined (Fig. 2a). However, samples with high tertile HLA-DR expression in the intraepithelial segments showed significantly longer RFS (Fig. 2b) compared to mid and low tertile samples. HLA-DR was more abundant in the stroma (Supplementary Fig. 1 and Supplementary Data 6), but high-level HLA-DR expression in stromal segments was not associated with better outcomes (Fig. 2c). Consistent with the data shown in Fig. 2, high intraepithelial HLA-DR expression was associated with improved outcome in all intraepithelial segment classes, CD45EnR, CD68EnR, and even lymphocyte-poor TumorEnR segments (Supplementary Fig 6a–c, respectively), whereas high-level HLA-DR expression in CD45EnR and CD68EnR stromal segments had no impact on survival t (Supplementary Fig. 6d, e).

In contrast to HLA-DR, IDO1 protein abundance was associated with improved outcomes in All Segments (combined intraepithelial + stromal) (Fig. 2d) as well as All Intraepithelial (Fig. 2e) and All Stroma (Fig. 2f) segments. Also, unlike HLA-DR, this association was observed in all CD45EnR, CD68ENR, and even in lymphocyte-low TumorEnR segments (Supplementary Fig. 7a–c, respectively) and both CD45EnR and CD68EnR stromal segments (Supplementary Fig. 7d, e). The data suggest that the spatial profiling capability of DSP analysis can develop a more detailed understanding of the link between clinical outcome and the intraepithelial and stromal immune architecture of specific immune protein biomarkers in TNBC.

Univariable Cox hazard ratios (HRs) were calculated for 10 differentially expressed proteins plus PD-L1. The data in Fig. 1 indicate that favorable outcome is associated with a subset of tumors with high intraepithelial protein expression, so we calculated HRs for tumors with upper tertile expression of each target protein vs. lower 2/3 tertile expression (Table 2). By univariable analysis, intraepithelial CD11c, CD20, CD40, CD56, HLA-DR, ICOS, and IDO1 were significantly associated with improved RFS (HRs, ranging from 0.2 to 0.3, $p < 0.05$). In stroma segments, only B2M (HR: 0.4), CD40 (HR: 0.3), and IDO1 (HR: 0.2) high-level expression was associated with RFS in univariable analysis.

Multivariable Cox HRs were calculated for 11 proteins comparing RFS in the top tertile to that in the bottom 2/3 with stromal TILs as a covariable (Table 2). Significant multivariable HRs were observed for intraepithelial CD20, CD40, CD56, GZMB, HLA-DR, and ICOS (HRs ranging from 0.2 to 0.3 at $p < 0.05$). Only GZMB was significant in the multivariable analysis of the stroma segments with sTILs as a covariable.

**Digital spatial profiling in the Mayo Clinic TNBC cohort**
Using a TMA constructed from the Mayo Clinic early-stage TNBC cohort, comprehensive clinicopathologic characterization, including recurrence data (invasive disease-free survival, IDFS), was available for

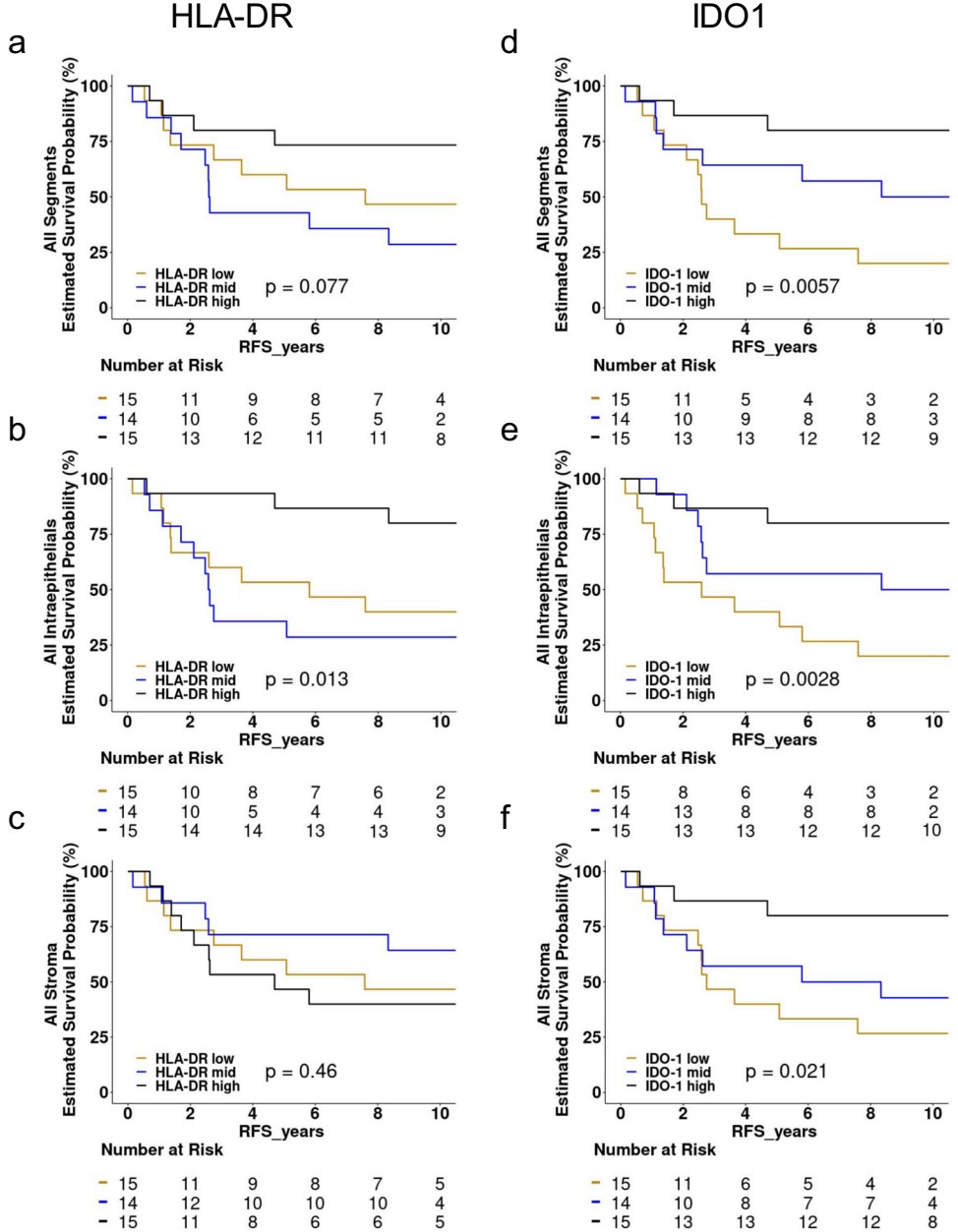

**Fig. 2 | Kaplan–Meier analysis of recurrence as a function of HLA-DR or IDO1 abundance in All Intraepithelial and All Stromal segments.** Tumor-average protein counts were used to stratify samples into tertiles (high = black, mid = blue, low = orange) based on the abundance of HLA-DR or IDO1. Kaplan–Meier analysis was carried out using protein abundance and time to event as continuous variables. Log-rank $p$-values are given, with the number of patients "at risk" shown below each curve. Source data are given in Supplementary Data 2. KM analysis of recurrence as a function of PD-L1 or IDO1 in CD45EnR Intraepithelial, CD68EnR Intraepithelial, TumorEnR Intraepithelial, CD45EnR Stroma, and CD68EnR Stoma segment classes are shown in Supplementary Figs. 6 (HLA-DR) and 7 (IDO1). HLA-DR abundance is given in All Segments (**a**), Intraepithelial segments (**b**), and Stromal segments (**c**); whereas IDO1 abundance is given in All Segments (**d**), Intraepithelial segments (**e**), and Stromal segments (**f**).

275 tumors that also had DSP data. In this cohort subset, approximately 53% of patients were post-menopausal, and most tumors were pT1-T2 (94%), pN0-1 (84%), and grade 3 (92%). Among the histologic subtypes, invasive ductal carcinomas (IDC) of no special type comprised 65% of tumors, followed by IDC with medullary features (21%), with 45% of the whole group having stromal TIL scores >20% (Supplementary Data 4). Unlike FinXX samples, the TMA was analyzed on a subsequently released commercial version of the GeoMx™ platform using a somewhat modified reagent set ("Methods"). Moreover, FinXX tumors were selected based on well-defined trial inclusion features, whereas the TMA included all evaluable TNBC tissue cores. Finally, FinXX data were from full-face tumor sections, whereas the TMA was

constructed from selected tissue cores. Despite these differences, normalized expression of the 31 proteins common to both experiments showed very high correlations within intraepithelial segments (Spearman's rho = 0.90) (Supplementary Fig. 8). The distribution of immune proteins between stromal and intraepithelial segments in the TMA samples (Supplementary Fig. 9 and Supplementary Data 6) was similar to that observed in FinXX (Supplementary Fig. 1). Similar subsets of immune proteins, including CD11c, CD40, HLA-DR, IDO1, PD-L1, and PD-L2, showed significant differences in protein abundance (log 2FC > 0.5, $p$ < 0.05) as a function of outcome in intraepithelial (Supplementary Fig. 10) or stromal (Supplementary Fig. 11) segments. Differential expression of 11 key immune proteins, expressed as log2 fold

**Table 2 | Cox hazard ratios (HRs) for recurrence in FinXX samples**

| Protein | Intraepithelial univariable HR (95% CI) | p-value | Stroma univariable HR (95% CI) | p-value | Intraepithelial multi-variable HR (95% CI) | p-value | Stroma multivariable HR (95% CI) | p-value |
|---|---|---|---|---|---|---|---|---|
| B2M | 0.458 (0.169, 1.244) | 0.116 | 0.338 (0.114, 1.000) | **0.040** | 0.532 (0.194, 1.456) | 0.219 | 0.415 (0.138, 1.247) | 0.117 |
| CD11c | 0.321 (0.109, 0.951) | **0.031** | 0.787 (0.321, 1.932) | 0.601 | 0.402 (0.133, 1.212) | 0.106 | 1.395 (0.544, 3.578) | 0.488 |
| CD20 | 0.318 (0.108, 0.942) | **0.029** | 0.817 (0.333, 2.005) | 0.658 | 0.283 (0.095, 0.839) | **0.023** | 0.718 (0.292, 1.768) | 0.471 |
| CD40 | 0.210 (0.062, 0.713) | **0.006** | 0.321 (0.109, 0.951) | **0.031** | 0.278 (0.081, 0.953) | **0.042** | 0.570 (0.179, 1.817) | 0.342 |
| CD56 | 0.209 (0.062, 0.710) | **0.006** | 0.452 (0.166, 1.226) | 0.109 | 0.206 (0.061, 0.701) | **0.011** | 0.473 (0.174, 1.285) | 0.142 |
| GZMB | 0.430 (0.158, 1.167) | 0.088 | 0.465 (0.171, 1.261) | 0.123 | 0.179 (0.062, 0.517) | **0.001** | 0.168 (0.058, 0.490) | **0.001** |
| HLA-DR | 0.204 (0.060, 0.695) | **0.005** | 1.488 (0.634, 3.490) | 0.358 | 0.263 (0.077, 0.905) | **0.034** | 1.628 (0.692, 3.829) | 0.264 |
| ICOS | 0.208 (0.061, 0.704) | **0.005** | 0.644 (0.252, 1.646) | 0.354 | 0.229 (0.067, 0.779) | **0.018** | 0.836 (0.321, 2.176) | 0.713 |
| IDO1 | 0.224 (0.066, 0.759) | **0.009** | 0.224 (0.066, 0.759) | **0.009** | 0.358 (0.099, 1.292) | 0.117 | 0.439 (0.092, 2.093) | 0.302 |
| PD-L1 | 0.610 (0.239, 1.560) | 0.297 | 0.821 (0.334, 2.014) | 0.666 | 0.530 (0.206, 1.359) | 0.186 | 0.936 (0.379, 2.314) | 0.886 |
| PD-L2 | 0.914 (0.373, 2.243) | 0.845 | 0.691 (0.270, 1.766) | 0.437 | 0.660 (0.266, 1.637) | 0.370 | 0.561 (0.218, 1.442) | 0.230 |

HRs were calculated using protein abundance (All Intraepithelial or All Stroma, as defined in the legend in Table 1) and time to event as continuous variables. Univariable and multivariable (adjusted for sTILs) HRs and p-values for 11 proteins (10 differentially expressed at BYadj p < 0.05 plus PD-L1). Bold font indicates two-sided p < 0.05. p-values were adjusted for stromal tumor-infiltrating lymphocytes (sTILs). Source data are given in Supplementary Table 2.

**Table 3 | Cox hazard ratios (HRs) for recurrence in Mayo Clinic TNBC_TMA samples**

| Protein | Intraepithelial univariable HR (95% CI) | p-value | Stroma univariable HR (95% CI) | p-value | Intraepithelial multi-variable HR (95% CI) | p-value | Stroma multivariable HR (95% CI) | p-value |
|---|---|---|---|---|---|---|---|---|
| B2M | 0.748 (0.466, 1.202) | 0.23 | 0.668 (0.412, 1.082) | 0.10 | 0.768 (0.468, 1.260) | 0.30 | 0.758 (0.459, 1.251) | 0.28 |
| CD11c | 0.638 (0.376, 1.084) | 0.10 | 0.884 (0.550, 1.423) | 0.61 | 0.565 (0.322, 0.994) | **0.05** | 0.752 (0.460, 1.230) | 0.26 |
| CD20 | 0.981 (0.617, 1.559) | 0.93 | 0.823 (0.509, 1.330) | 0.43 | 0.961 (0.588, 1.569) | 0.87 | 0.808 (0.492, 1.328) | 0.40 |
| CD40 | 0.458 (0.257, 0.814) | **0.01** | 0.568 (0.340, 0.949) | **0.03** | 0.513 (0.278, 0.944) | **0.03** | 0.702 (0.407, 1.210) | 0.20 |
| CD56 | 0.724 (0.443, 1.182) | 0.20 | 0.883 (0.560, 1.393) | 0.59 | 0.617 (0.371, 1.026) | 0.06 | 0.816 (0.515, 1.295) | 0.39 |
| GZMB | 1.230 (0.794, 1.904) | 0.35 | 0.904 (0.577, 1.415) | 0.66 | 1.091 (0.676, 1.762) | 0.72 | 0.823 (0.513, 1.320) | 0.42 |
| HLA-DR | 0.472 (0.270, 0.824) | **0.01** | 0.905 (0.570, 1.439) | 0.67 | 0.509 (0.285, 0.909) | **0.02** | 1.059 (0.658, 1.705) | 0.81 |
| ICOS | 0.599 (0.357, 1.003) | **0.05** | 0.507 (0.295, 0.872) | **0.01** | 0.759 (0.436, 1.321) | 0.33 | 0.611 (0.349, 1.071) | 0.09 |
| IDO1 | 0.534 (0.318, 0.894) | **0.02** | 0.618 (0.382, 1.000) | **0.05** | 0.731 (0.417, 1.280) | 0.27 | 0.816 (0.487, 1.369) | 0.44 |
| PD-L1 | 0.543 (0.315, 0.935) | **0.03** | 0.514 (0.303, 0.873) | **0.01** | 0.664 (0.372, 1.185) | 0.17 | 0.657 (0.373, 1.157) | 0.15 |
| PD-L2 | 0.866 (0.540, 1.390) | 0.55 | 1.314 (0.863, 2.002) | 0.20 | 0.926 (0.567, 1.512) | 0.76 | 1.331 (0.861, 2.055) | 0.20 |

Univariable and multivariable HRs were calculated for Mayo Clinic TNBC patients using time to event as a continuous variable and protein abundance in All Intraepithelial (n = 12/samples) and All Stroma (n = 8/sample) segments, as defined in the legend to Table 1. In multivariable analysis, p-values were adjusted for known clinical variables, including patient age at surgery, tumor size and grade, lymph node status, and sTIL scores. Bold font indicates two-sided p < 0.05. Source data are given in Supplementary Data 7.

change (FC) for IDFS = YES vs. IDFS = NO, was similar in intraepithelial segments from both FinXX and Mayo Clinic TNBC samples (Supplementary Table 2).

**Survival analysis in the Mayo Clinic TNBC cohort**

By univariable analysis, intraepithelial enrichment of CD40, HLA-DR, ICOS, IDO1, and PD-L1 was associated with better outcomes in the Mayo Clinic TNBC TMA cohort (Table 3), similar to findings with the FinXX samples (Table 2). Stromal CD40, IDO1, ICOS, and PD-L1 were also associated with improved DFS in univariable analysis. By multivariable analysis, adjusting for patient age at surgery, tumor size and grade, lymph node status, and sTIL scores, high levels of intraepithelial CD40 [HR: 0.51 (0.28, 0.9), p = 0.032], HLA-DR [HR: 0.51 (0.29, 0.91), p = 0.023], and CD11c [HR: 0.57 (0.32, 0.99), p = 0.048] were associated with improved HR, with CD56 falling just short of statistical significance [HR: 0.62 (0.37, 1.03), p = 0.063]. No immune protein in the stromal segments showed a significant association with IDFS by multivariable analysis (Table 3). Together, these data suggest that a set of intraepithelial antigen-presentation immune proteins are prognostic in treatment-naive TNBC, independently of sTILs and other established clinicopathologic variables.

The high multiplexed and quantitative nature of DSP data facilitates the calculation of eigenprotein scores, based on the scaled abundance of sets of functionally related proteins designed to reflect the activity of the cognate function. We used a scaled abundance of HLA-DR, CD11c, and CD40 to calculate an eigenprotein score of intraepithelial expression of antigen-presenting cell proteins, the APC score (APC scores for all samples are given in Supplementary Data 5). The univariable Cox HR for the APC score was 0.56 (95% CI: 0.34, 0.94, p = 0.020) (Supplementary Fig. 12), and the log-rank p-value for Kaplan−Meier analysis of APC > median vs. APC < median was 0.003 (Supplementary Fig. 13), indicating that the APC score has strong prognostic significance. For reference, we calculated the Cox HR for stromal TILs (HR = 0.97, 95% CI 0.94, 0.99, p = 0.01, Supplementary Fig. 12). ROC analysis indicated that the APC score is not inferior to stromal TILs at prognosticating outcome (Supplementary Fig. 14) when both features are analyzed as continuous variables. A second eigenprotein score was calculated, using the scaled abundance of CD3 and ICOS, to reflect the intraepithelial T-cell activation status (TCA score, Supplementary Data 5) of each tumor (Cox HR 0.63, 95% CI: 0.36,1.08, p = 0.08, Supplementary Fig. 12).

Intraepithelial APC and TCA scores were highly correlated in both FinXX and TNBC samples (Supplementary Fig. 15). Samples with high (>median) scores for both features (APChigh/TCAhigh) exhibited a significantly reduced risk of recurrence, compared to samples with low (<median) scores for both APC and TCA scores (FinXX OR 0.24, p = 0.05; TNBC_TMA OR 0.46, p = 0.02). Kaplan−Meier analyses of APChigh/TCAhigh vs. APClow/TCAlow are shown in Supplementary

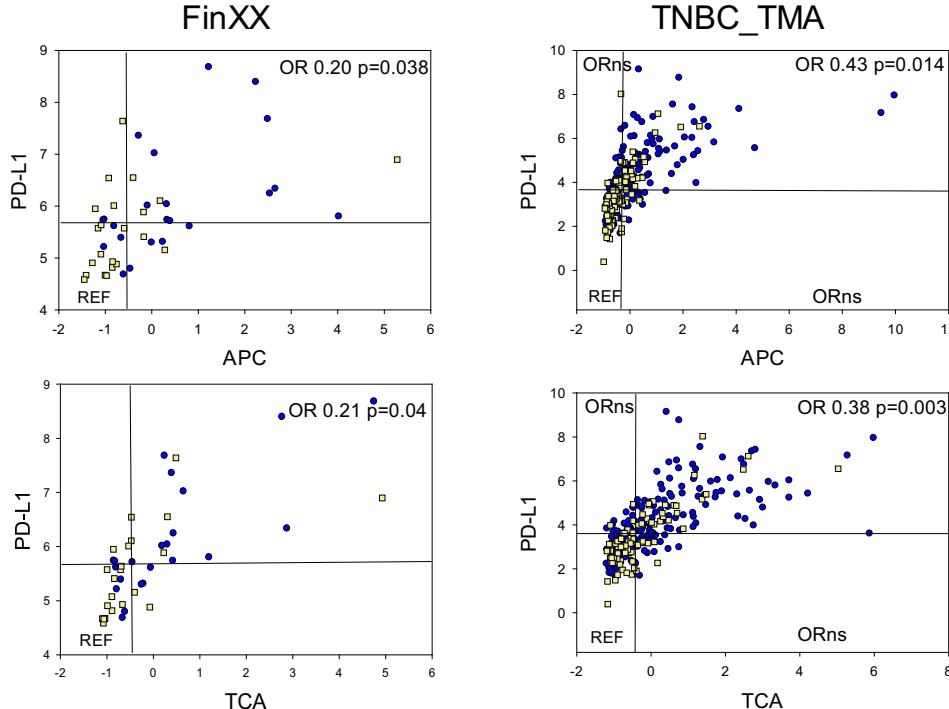

**Fig. 3 | Recurrence in FinXX and Mayo Clinic TNBC_TMA cohorts as a function of PD-L1 protein abundance and the APC or TCA eigenprotein scores.** Each dot or square represents tumor-average values for each feature in each tumor, including tumors that recurred (yellow squares) and tumors that did not recur (blue circles). The $X$ axis contains numerical scores for the APC or TCA eigenproteins, and the $Y$ axis contains PD-L1 protein log2 normalized counts for intraepithelial segments. Odds ratios (ORs) for 5-year recurrence were calculated as dichotomous variables (recurrence YES or NO). The reference lines indicate the population median values for each feature, and ORs were calculated in reference to samples that were

<median for both features (e.g., APClow/TCAlow). Significance was defined as $p < 0.05$, and "quadrants" with $p > 0.05$ for ORs are designated as "ORns". ORs for FinXX APClow/TCAhigh and APChigh/TCAlow are not given in Fig. 3, considering the low number of samples in these quadrants. However, ORs for all quadrants are given in Supplementary Table 3. Source data are given in Supplementary Data 2 (FinXX tumor-average abundance), Supplementary Data 7 (TNBC_TMA), and Supplementary Data 5 (eigenprotein scores). Patient numbers: FinXX, 22 recurred and 22 did not recur; TNBC_TMA, 81 recurred and 181 did not recur.

Fig. 16 (FinXX: log-rank $p = 0.002$ and TNBC TMA: log-rank $p = 0.02$ for APChigh/TCAhigh vs. APClow/TCAlow). Odds ratios for 5-year recurrence are given in Supplementary Table 3. Of note, the small set of samples which were APChigh and TCAlow or vice versa did not have a significant reduction in risk of recurrence, compared to APClow/TCA-low samples. This observation is consistent with the hypothesis that co-localized antigen presentation and T-cell function are necessary for clinically impactful immune-mediated tumor destruction.

We evaluated the relationship between intraepithelial PD-L1 protein, a clinically actionable immune checkpoint protein in TNBC, and APC or TCA immune scores. PD-L1 is expressed by professional antigen presenters such as macrophages and dendritic cells, but also lymphocytes, and more rarely, tumor cells in TNBC; it remains unclear which PD-L1-expressing cell populations, and their spatial microenvironment, contribute to the clinical or therapeutic benefit seen in PD-L1 positive TNBC. Tumors with high APC scores and high (>median) intraepithelial PD-L1 abundance exhibited significantly better prognosis than APClow/PD-L1low tumors (FinXX: OR 0.20, $p = 0.038$; TNBC_TMA: OR 0.43, $p = 0.014$, Fig. 3 and Supplementary Table 3). KM curves for APChigh/PD-L1high vs. APClow/PD-L1low are shown in Supplementary Fig. 17. However, PD-L1high TNBC with low APC scores did not exhibit a significantly reduced risk of recurrence. Note in Fig. 3 that 4/5 APClow/PD-L1high FinXX tumors recurred (yellow squares), and the ORs for 5-year recurrence (Supplementary Table 3) were not significantly different in APClow/PD-L1high and APClow/PD-L1low. Similar to the data with APC, TCAhigh/PD-L1high tumors had a significantly reduced risk of recurrence compared to TCAlow/PD-L1low TNBC (Fig. 3: FinXX OR 0.21, $p = 0.004$; TNBC TMA OR 0.38, $p = 0.003$).

Conversely, TCAlow/PD-L1high tumors did not exhibit a significantly better prognosis than TCAlow/PD-L1low tumors. Note in Fig. 3 that 4/7 FinXX tumors with TCAlow aPD-L1high recurred (yellow squares), and the ORs for 5-year recurrence in these TCAlow/PD-L1high tumors in both tumor cohorts were not significantly different from that observed in TCAlow/PD-L1low tumors (Supplementary Table 3). Together, these data indicate that among TNBC with high intraepithelial PD-L1 protein, concurrent high antigen presentation function or T-cell activation (or, more likely, both) are required to effect significant reductions in the risk of tumor recurrence.

IDO1, another immune checkpoint protein, was significantly associated with durable RFS in both univariable and multivariable analyses of the FinXX samples (Table 2). Tumors with high IDO1 and high APC or TCA scores (Fig. 4) exhibited a significantly reduced risk of recurrence within 5 years (APChigh/IDO1high vs. APClow/IDO1low in FinXX: OR 0.17, $p = 0.02$; TNBC TMA: APChigh/IDO1high vs. APClow/IDO1low, OR 0.44, $p = 0.019$; APClow/IDO1high vs. APClow/IDO1low OR: 0.27, $p = 0.014$; TCAhigh/IDO1high vs. TCAlow/IDO1low in FinXX: OR 0.09, $p = 0.01$; TNBC TMA: TCAhigh/IDO1high vs. TCAlow/IDO1low, OR 0.39, $p = 0.005$, Supplementary Table 3). KM curves are shown in Supplementary Fig. 18. In contrast to PD-L1, a significant reduction in recurrence risk was observed in intraepithelial APClow/IDO1high tumors. In the FinXX samples, only 1/4 APClow/IDO1high tumors recurred (yellow squares in Fig. 4), and the OR for 5-year recurrence in the Mayo TNBC cohort was 0.27, $p = 0.014$ (Supplementary Table 3). However, high IDO1 was not associated with a statistically significant reduction in the risk of recurrence in tumors with low TCA scores, Fig. 4).

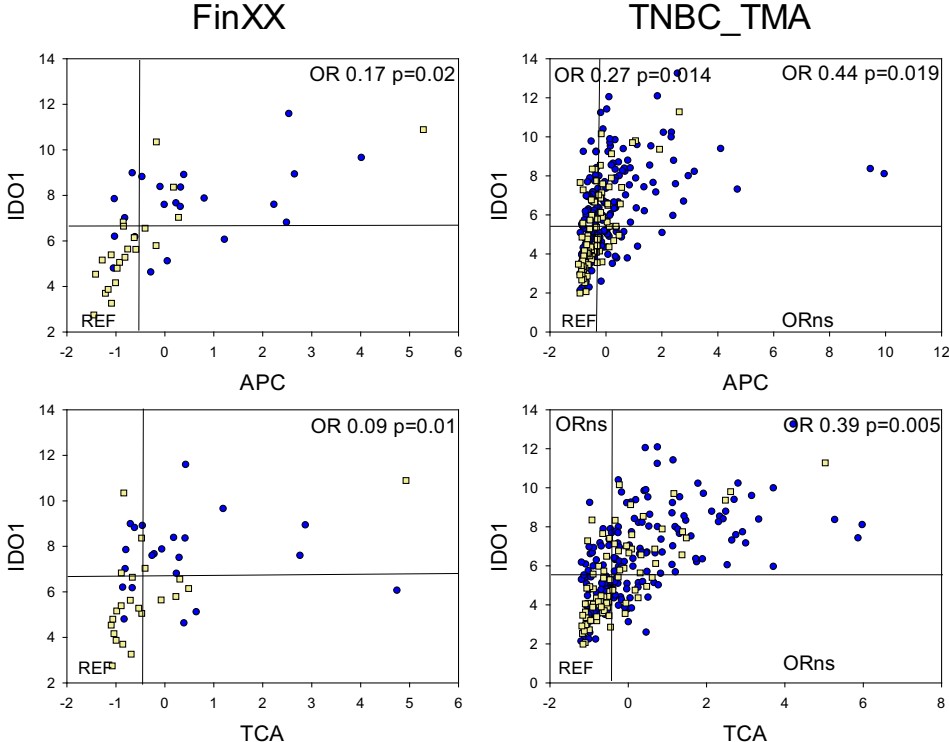

**Fig. 4 | Recurrence in FinXX (left panel) and Mayo Clinic TNBC_TMA (right panel) cohorts as a function of IDO1 protein abundance and APC or TCA eigenprotein scores.** As in Fig. 3, each dot or square represents an individual tumor, including tumors that recurred (yellow squares), and tumors that did not recur (blue circles). Numerical values for APC or TCA eigenprotein scores are given on the X axis, and normalized log2 counts for IDO1 protein are given on the Y axis. ORs were calculated for 5-year recurrence. Reference lines indicate the median values for each feature, and ORs were calculated in reference to samples that were <median for IDO1 and APC/TCA scores. ORs with $p < 0.05$ are designated "ORns", and ORs for all segments are given in Supplementary Table 8. Source data for eigenprotein scores are given in Supplementary Data 5. Source data for protein abundance are given in Supplementary Data 2 (FinXX) and Supplementary Data 7 (TNBC_TMA). Patient numbers: FinXX, 22 recurred and 22 did not recur; TNBC_TMA, 81 recurred and 181 did not recur.

## Discussion

Intrinsic host-tumor immune responses have established prognostic and predictive roles in patients with TNBC. Beyond immunotherapeutic strategies in advanced disease, immune biomarkers can potentially inform tailored therapeutic strategies in patients with early-stage or advanced TNBC[28,29]. Although stromal TIL scores have established prognostic value, they serve as a surrogate of a complex interplay of immune cell populations, cannot provide biological or clinically relevant insights into the immune activation status of the stromal immune milieu, and do not directly elucidate the immune landscape of the intraepithelial compartment. Ultimately, however, the ability to reduce breast cancer mortality depends upon a fundamental understanding of the biology that underlies tumor cell/immune cell interaction, and to a substantial extent, the studies described above focus on providing an in-depth analysis of the relationship between clinical outcome (recurrence) and the immune landscape of tumor cells and their microenvironment. NanoString GeoMx™ technology is a powerful tool for measuring the abundance of immune and other key proteins in both stromal and intraepithelial compartments, and this technology has been applied to breast cancers[14,30], prostate cancers[31], lung cancers[32], and melanoma[25], among others. We have used the GeoMx™ digital spatial profiling platform to characterize immune microlandscapes in two cohorts of early-stage systemic treatment-naïve TNBC to quantify spatially defined immunoproteins of intrinsic TIME and inform the immunobiology and clinical utility of established and emerging clinically actionable biomarkers in TNBC.

Generally, the stromal segments had higher levels of almost all immune proteins relative to the intraepithelial compartment, irrespective of its enrichment in CD45/CD68+ cells (FinXX) or with random selection (TNBC TMA). Nevertheless, the immune milieu of the adjacent stroma did partially inform the adjacent intraepithelial tumors segment, as evidenced by the observation that CD45+ stroma-enriched intraepithelial tumor segments differed from the other two classes in FinXX. Moreover, intraepithelial TIMEs maintained both qualitative and quantitative differences in immune protein architecture from stroma segments, across all segment classes. For example, the amount of HLA-DR in intraepithelial segments was not impacted by the immune milieu of the adjacent stroma (CD68EnR vs. CD45EnR cell enrichment). However, the amount of intraepithelial HLA-DR was significantly higher in CD45EnR or CD68EnR epithelial segments than that observed in TumorEnR intraepithelial segments with no adjacent immune-enriched stroma.

Arguably, more importantly, we observed that the quantity of a select set of immune proteins in the intraepithelial segments, rather than stromal segments drove associations with durable RFS. For example, while HLA-DR protein was more abundant in stroma than in tumor (intraepithelial segments), only intraepithelial HLA-DR abundance was significantly associated with outcome. Moreover, by KM analysis, intraepithelial HLA-DR protein enrichment (highest tertile vs. lower two tertiles) was associated with RFS in all three intraepithelial segment types in FInXX tumors, but in no stromal segment type. This observation suggests that stromal immune architecture may not reflect or inform the prognostic antigen-presentation features of intraepithelial segments.

Specifically, we identified an intraepithelial immune protein set enriched in proteins associated with antigen presentation, including CD11cCD40, and HLA-DR. HLA-DR is one component of the MHC class

II protein complex, typically expressed by professional antigen-presenting cells, and expression of genes comprising class II MHC are associated with outcome in TNBC r[33,34]. In both the FinXX and Mayo Clinic TNBC cohorts, tumors with high intraepithelial HLA-DR had significantly improved clinical outcomes, independent of stromal TIL scores and other established clinicopathologic variables. A prior small study of 10 TNBC tumors also demonstrated that intraepithelial HLA-DR levels were higher in tumors of patients with better clinical outcomes[30] and verified that HLA-DR abundance by DSP was highly correlated with semi-quantitative immunohistochemistry scores of HLA-DR. However, to date, HLA-DR protein studies have been limited by study size or semi-quantitative immunohistochemical techniques and have shown no independent association of HLA-DR with outcomes in TNBC[35,36]. One important biologic question is whether the tumor or immune cells, or both, are expressing HLA-DR. Although DSP cannot directly determine which cell type is expressing a target protein, a prior study has demonstrated that both tumor cells and immune cells can express components of the MHC II pathway, and likely both facilitate antitumor immunity in TNBC[34]. Multivariable analysis of the TNBC TMA cohort confirmed that intraepithelial HLA-DR, CD11c, and CD40 were prognosticators of recurrence, independent of TILs and other established clinicopathologic prognostic variables. CD40, a co-stimulatory transmembrane protein within the tumor necrosis factor superfamily, is a pan-activator of antigen-presenting cells. An analysis of publicly available data sets of bulk gene expression data has shown that splice variants of CD40 are associated with outcomes in TNBC patients;[37] however, there are limited data on CD40 protein expression in TNBC tumoral tissue, with one study reporting that cytoplasmic immunohistochemical staining of CD40 in breast cancer (not TNBC specifically) was associated with outcome[38]. Nevertheless, CD40 is an emerging therapeutic target in multiple cancer types: CD40 agonist antibodies, combined with a T-cell-inducing vaccine and PD-1 antagonist antibodies, showed promising antitumor activity in an orthotopic breast cancer model[39], and several clinical trials combining agonist CD40 antibodies with various (immuno)therapies for TNBC and other cancer types are ongoing. CD11c was also independently associated with outcomes in the TNBC TMA cohort. CD11c-positive cells have been associated with increased TILs[40] and favorable outcomes in TNBC[41]. Based on these observations, we used the high-plex nature of the data to develop an eigenprotein score that consolidated the scaled abundance of intraepithelial HLA-DR, CD11c, and CD40 proteins, the APC score, as a surrogate marker of antigen-presenting activity within the intraepithelial segments of FinXX and TNBC TMA tumors. The digital nature of the GeoMx DSP data facilitates the generation of such aggregate scores, which would be difficult to develop using analog data, such as conventional immunohistochemistry. The intraepithelial APC score was strongly associated with a reduced risk of recurrence.

We generated a second eigenprotein score using the intraepithelial abundance of ICOS and the T-cell receptor CD3 as a putative marker for T-cell activation status. ICOS is a marker of T-cell activation, and intraepithelial ICOS was associated with favorable outcomes in univariable analysis and independent of TILs in the FinXX cohort. Notably, the intraepithelial TCA score alone was not significantly associated with recurrence risk (Cox HR $p = 0.08$). Only TNBC with high (>median) scores for both APC and TCA exhibited a significantly reduced risk of recurrence in both TNBC cohorts. It is perhaps unsurprising that TNBC tumors with high levels of antigen-presenting and T-cell activation in the intraepithelial compartments exhibit better prognosis. Nevertheless, a direct demonstration of this relationship provides a foundation for the development of these features as potential spatial biomarkers for both prognosis and prediction. The quantitative (digital) nature of the GeoMx data provides a dynamic range and degree of precision that is difficult to obtain with conventional histopathological biomarkers, and the fact that the individual components of these scores are significantly associated with outcome

independent of TILs suggests a significant potential for the development of spatial prognostic and/or predictive models focused on immune features that reflect direct interaction between tumor and immune cells. Taken together, these observations support the hypothesis that the immune cells engaged in intimate contact with the tumor cells may be as relevant, perhaps even more relevant, to the clinical outcomes than the unengaged immune cells in the adjacent stroma.

PD-L1 is a well-known, albeit poorly understood, prognostic and predictive indicator in subsets of TNBC[42,43]. Antibodies that target the PD-(L)1 axis are approved for both early- and late-stage TNBC, although for early-stage TNBC, the relationship between PD-L1 expression (defined by the 22C3 antibody companion diagnostic assay) and response to anti-PD1 (pembrolizumab) is obscure[4]. We have taken advantage of the quantitative spatial nature of the GeoMx data to interrogate the relationship between clinical outcome, intraepithelial PD-L1 abundance, and intraepithelial immune features such as antigen-presenting (APC) or T-cell activation (TCA) status.

The data indicate that PD-L1 is associated with favorable outcomes only in TNBC with high levels of antigen-presenting and/or T-cell activation status. Tumors with low APC or TCA scores did not have improved outcome with concurrent high intraepithelial PD-L1 expression. This conclusion may inform the observation that PD-L1 protein abundance with the immunohistochemical 22C3 companion diagnostic assay does not appear to predict clinical response to pembrolizumab in early-stage TNBC[4], and it is tempting to speculate that APClow/PD-L1high or TCAlow/PD-L1high tumors will not respond to anti-PD1 therapy. While these data do not directly address the predictive potential of intraepithelial APC and or TCA scores as predictors of response to immune checkpoint inhibitor therapy, experiments to evaluate such relationships are ongoing.

In both TNBC cohorts, tumors with IDO1 enrichment in intraepithelial or stromal compartments were associated with improved clinical outcomes (by univariate analysis). IDO1 (Indoleamine 2,3-dioxygenase), an enzyme involved in tryptophan catabolism, functions as an immune checkpoint molecule and is correlated with sTILs[44]. We have previously shown that IDO1 is co-expressed with PD-L1 in early-stage TNBC[27]. IDO1 provides an interesting contrast to PD-L1. TNBC with high intraepithelial IDO1 tended to have a better prognosis, irrespective of antigen-presenting activity (APC score). Moreover, although high APC scores generally reflected durable RFS or IDFS, tumors with high APC but low IDO1 tended to have poor prognoses. Tumors with high IDO1 but low T-cell activation status (TCA scores) also had poor prognoses, consistent with the conclusion that the interaction between IDO1 expression and T-cell activation is associated with favorable outcomes. Unlike HLA-DR, both PD-L1 and IDO1 are conventionally associated with immunosuppressive roles, and the paradoxical observation that high intraepithelial PD-L1 and IDO1 are associated with improved outcomes in early-stage TNBC appears counterintuitive. However, similar associations of IDO1 and PD-L1 with outcome or pathologic complete response in the neoadjuvant setting, have been observed with bulk transcriptomic profiling in TNBC[45,46] and by conventional immunohistochemistry[47] and in other tumor types[48]. We suspect that the nuances of which specific cell populations express PD-L1 or IDO1 in TNBC (neoplastic cells, immune cells, or endothelial/non-immune stromal cells), their spatial context, and accompanying neighbor cell–cell interactions may underlie this apparent paradox. Emerging high-plex spatial platforms with single cell resolution should offer insight into these questions and we have efforts underway to address these questions with new technology. In short, these quantitative spatial analyses provide fundamental insight into the intrinsic biology of the immune micro-architecture in treatment-naïve TNBC, in two large cohorts of well-annotated early-stage TNBC. Such analyses may inform therapeutic decision-making in the setting of advanced disease with promising preclinical or early clinical data for drugs that

target CD40[39], IDO1, and HLA-DR, either alone or in combination with immune checkpoint blockade in carcinomas or melanoma[30, 49].

High-plex digital spatial profiling with the GeoMx™ platform has a wide dynamic range, high reproducibility, and high correlation with protein quantitation by other methods[23, 26]. In our study, we also observed high reproducibility of normalized protein levels across the two TNBC cohorts, across a wide spectrum of low- and high-abundance proteins. This assay sensitivity is of potential clinical utility as multiple immune checkpoint proteins, including PD-L1, are often low-abundance proteins in TNBC and have very low clinically actionable cut-points[3]. However, the NanoString GeoMx™ platform does have its limitations. Although micron-scale microenvironments can be interrogated with quantitative protein data, neither single cell resolution nor protein co-expression in the same cell can be determined. Nevertheless, the high sensitivity, high resolution, and spatial capabilities of the platform offer heretofore unseen biologic and clinically relevant insights into the spatially defined immune protein architecture. Another study limitation is that the FinXX and Mayo Clinic treatment-naïve TNBC cohorts differed substantially in sample number, patient selection, and treatment protocols, and used beta and commercial versions of the DSP equipment and reagents, respectively; additional study is required to define clinically relevant immune protein cut-points for these spatially constrained biomarkers.

In summary, in two independent cohorts of treatment-naïve TNBC, we mapped and quantitated dozens of immune proteins and determined that spatial context influences immune microenvironments, with substantive differences between intraepithelial segments and adjacent stromal immune microenvironments. Intraepithelial antigen presentation-related immune proteins(HLA-DR, CD11c, and CD40), are independently prognostic of clinical outcomes, and functional antigen presentation and T-cell activation eigenprotein scores inform the prognostic context of clinically actionable biomarkers (e.g., PD-L1). These data provide insight into immune-based therapies and their companion diagnostic assays and complement existing immune-based prognosticators in TNBC, such as sTILs, to optimize patient selection for immune-based or other therapies from these.

## Methods

This study was conducted in accordance with recognized ethical guidelines, including the U.S. Common Rule. The FinXX study was approved by an Institutional Review Board at the Helsinki University Hospital (approvals 264/13/03/02/2014 and HUS/903/2017). The patients who participated in the FinXX trial (NCT00114816) signed a written informed consent to the trial participation and consent to allow the use of their tumor tissue for FinXX trial-related research purposes. Analysis of Mayo Clinic TNBC samples was approved by the Mayo Clinic Institutional Review Board. Written informed consent was obtained from all patients for the use of tumor samples for research purposes.

### Patient samples

Formalin-fixed, paraffin-embedded (FFPE) tumor tissues were available from 120 centrally evaluated, early-stage patients with TNBC, enrolled in the FinXX trial evaluating adjuvant fluoropyrimidine therapy (ClinicalTrials.gov identifier NCT00114816)[50]. Only female breast cancer patients were analyzed, with gender self-identified. Patients were randomized to two arms, receiving either 5-fluorouracil (5FU) or capecitabine (a 5FU pro-drug). Both drugs share a common mechanism of action, and similar outcomes were observed in the trial, so we elected to combine samples from both arms for our analyses. RFS was used for outcome analyses, as described in the original report of the FinXX trial[50].

From the FinXX trial, a case/control design was used to identify proteins that were differentially associated with the outcome. We selected all patients who had a recurrence (22 patients total, 11 from

each treatment arm) and 22 patients without recurrence (11 from each arm), matched for patient age, tumor grade, tumor size, and lymph node status for digital spatial profiling (DSP) of FFPE-derived tumor sections (Supplementary Table 1). For orthogonal analysis, we performed DSP in a TMA constructed from the Mayo Clinic TNBC cohort. This cohort is composed of 605 patients with early-stage, centrally verified TNBC (ER/PR < 1% and HER2-negative) who underwent surgery at Mayo Clinic (1985–2012). Tumor samples were collected prior to any systemic therapy. A detailed clinicopathologic review of the Mayo TNBC cohort has been previously published and the corresponding TMA is composed of 2–3 ×1 mm tissue cores from representative FFPE tumor blocks, as previously reported[51, 52]. Outcome analyses with the TNBC cohort were carried out using invasive disease-free survival (IDFS), as recommended in Hudis et al[53]., to conform to the outcome analyses reported in the original paper on this Mayo Clinic TNBC cohort[51]. Patient demographics for the included Mayo Clinic TNBC samples are given in Supplementary Data 4.

### Digital spatial profiling—selection and segmenting of regions of interest

A beta version of the GeoMx™ instrument and beta software was used to analyze the FinXX samples (Mayo Clinic Florida). Full-face FFPE tumor sections were stained with a cocktail of fluorophore-tagged antibodies against pan-cytokeratin (CK), CD45, and CD68 plus SYTO13 nuclear dye. The cocktail also contained 45 target or control antibodies, each tagged via a UV-labile cross-linker with a synthetic oligonucleotide bar code. Stained slides were scanned using the GeoMx™ beta instrument, and 4-channel immunofluorescent (IF) digital images (Supplementary Fig. 19) were generated to visualize tumor cells (CK+), leukocytes (CD45+), and macrophages (CD68+) plus nuclei (SYTO13). These images were used to select twelve 300-micron circular regions of interest (ROIs) per tumor. ROIs were targeted to regions of tumor nests with different stromal characteristics, including tumor-enriched areas with little or no associated stroma ("TumorEnR", N = 4), tumors with adjacent CD45-enriched stroma ("CD45EnR", N = 4), or tumor with adjacent CD68-enriched stroma ("CD68EnR", N = 4) (Supplementary Fig. 19a). The immunofluorescent images within each ROI were used to define computer-generated masks to guide laser illumination. Two such masks were generated for each ROI. The CK-positive mask defines the tumor nest, hereinafter referred to as the intraepithelial domain, which contains both CK-positive tumor cells and CK-negative immune and other cell types that have infiltrated the tumor nest. A CK-negative/SYTO13-positive mask was generated to accommodate the stromal compartment, which contains all nucleated CK-negative cells outside the tumor nest. These two masks define areas of illumination (AOIs) by the computer-guided laser. This process, known as segmentation, thus generates two distinct segments within each ROI, which will be referred to as the intraepithelial and stroma segments (Supplementary Fig. 19b–e). Intraepithelial segments were illuminated first to release antibody-bound oligonucleotide tags. These were collected first, followed by illumination and collection of antibody-bound oligonucleotides from the stromal segments. Oligonucleotides were quantitated with conventional NanoString nCounter technology. Data were normalized, according to NanoString "best practice" recommendations[24], to the geometric mean of two housekeeping proteins: histone H3 and ribosomal protein S6, using a beta version of the GeoMx™ software. Among 1050 segments, we excluded those with <30 nuclei or <1000 microns squared area, leaving 950 segments for analysis. The majority of stroma segments from TumorEnR ROIs failed QC due to low nuclear counts, and all TumorEnR stroma segments were deleted from further analysis.

Analysis of the Mayo Clinic TNBC cohort-derived TMA was performed with the commercial version of the GeoMx™ instrument (Mayo Clinic Florida) using GeoMx software version 1.3.0.20. As the evaluable region of a stained slide with the GeoMX instrument is smaller than the

size of the Mayo Clinic TNBC TMA slides, only a subset of tissue cores from the TMA was captured. Consequently, we did not have duplicate cores from all samples. Those with duplicate cores were averaged. A single 600-micron diameter circular ROI was selected from each 1 mm tissue core. No attempt was made to select ROIs based on the immune cell composition of the adjacent stroma. Consequently, the TMA samples represent a random selection of CD45, CD68, or tumor-enriched ROIs. ROI segmentation was identical to that described for FinXX samples (intraepithelial and stromal segments). Specifically, a total of 841 ROIs (from individual tissue cores) passed quality control for nuclear content and segment area: 416 intraepithelial segments and 425 stroma segments, corresponding to 275 unique tumors from the TNBC cohort with intraepithelial and stroma segment-level DSP data. The commercial antibody kit contained 62 oligonucleotide-tagged antibodies, of which 31 were common to the antibody kit used for the FinXX TNBC samples[54]. The data normalization strategy was identical to that used for FinXX TNBC samples.

### Antibodies

Forty-five oligonucleotide-tagged antibodies were included in the beta reagent kit used to analyze the FinXX samples (Supplementary Data 1), including three antibodies to housekeeping proteins (ribosomal protein S6, histone H3, and GAPDH), three negative controls (Mouse IgG1, Mouse IgG2a, Rabbit IgG) to estimate the lower limits of detection (defined as the mean counts from all negative controls plus two standard deviations) and one negative hybridization control. Validation of the antibodies listed in Supplementary Data 1 is described in Merritt et al[23]. Twenty-nine target proteins were above the lower limits of detection in the FinXX cohort and were included in the analysis.

Anti-CD68 (Novus NBP2-34736AF647, clone SPM130, 0.25 μg/μl), labeled with Alexa Fluor 647, was obtained from NovusBio. Anti-pan-cytokeratin labeled with Alexa Fluor 532 (Novus NBP2-33200AF532, clones A1+A3, lot 42604602, 0.5 μg/μl) and anti-CD45 labeled with Alexa Fluor 594 (Novus NBP2-34528AF594, clones 2B11+PD7/26, lot 42604402, 5.0 μg/μl) plus SYTO13 were obtained from NanoString. Antigen retrieval, antibody binding, and immunofluorescent staining were performed per NanoString protocol (https://blog.nanostring.com/GeoMx-online-user-manual/Content/Home2.0.htm).

### FinXX analysis

For segment-level analysis, we compared protein abundance in intraepithelial or stromal segments per ROI. For most analyses, intraepithelial or stromal segments were combined (e.g., All Intra-pithelial or All Stroma). If not, this is indicated by the ROI subtype (e.g., CD45EnR Stroma segments vs. CD45EnR intraepithelial tumor segments). For outcome analysis, tumor level abundance (rather than segment) was calculated as the mean of all segment-level protein counts. For example, for tumor level analysis, CD45EnR Tumor equals the mean of four CD45EnR segments for each tumor. Stromal tumor-infiltrating lymphocytes (TILs) were scored on digital images of the H&E-stained tumoral sections by a board-certified pathologist (JMC) using the recommendations from the International Immuno-oncology Biomarker working group[7]. Outcome analysis (Cox HRs) was calculated for the top tertile expression for individual proteins versus the bottom 2/3.

### Mayo Clinic TNBC cohort analysis

Only female breast cancer patients were analyzed, with gender self-identified. Segments from duplicate tumoral tissue cores were averaged (if >1), and differentially expressed proteins were identified using a negative binomial generalized linear model. The commercial antibody cocktail that was used for these samples contained 59 antibodies targeting three negative controls, three housekeeping proteins, one hybridization control, and 53 target proteins (Supplementary Table 7).

Our analyses focused only on those target proteins that were common to both the beta reagent set (FinXX) and the commercial reagents.

### Eigenprotein score calculation

Eigenprotein scores were calculated as the first principal component[55] of either CD11c, CD40, and HLA-DR scaled abundance (APC score) or ICOS plus CD3 scaled abundance (TCA score). Outcome analysis (OR or Cox HR) for eigenprotein score was calculated based on median score (high => median, low =≤ median),

### Statistical analysis

All secondary analyses were carried out using R Statistical Software (version 4.1.2; R Foundation for Statistical Computing, Vienna, Austria). For segment-level analysis, the linear mixed model was used to calculate the differential abundance of proteins as a function of spatial location (e.g., CD45EnR Tumor vs. CD45EnR Stroma). At the tumor level, segments were averaged, and the negative binomial generalized linear model was used for recurrence-free survival as a categorical variable (RFS = YES/NO). Differential abundance (listed as log fold change (FC) in all figures and tables) was estimated from the general linear model, and adjusted for multiple testing[56]. Differential expression [listed as log fold change (FC) in all Figs. and Tables] was estimated from the general linear model with significance defined as two-sided $p < 0.05$. Using target protein abundance, tumors were stratified into tertiles, and Kaplan–Meier survival analysis was performed for RFS as a continuous variable versus protein abundance. Statistical significance was defined as log-rank $p < 0.05$. Hazard ratios were calculated using the Cox model. In addition, we calculated odds ratios for 5-year recurrence (ORs) using a logistic regression model to analyze eigenprotein scores in combination with PD-L1 or IDO1 protein abundance.

### Reporting summary

Further information on research design is available in the Nature Portfolio Reporting Summary linked to this article.

## Data availability

All data used in this study are available as a supplement to the manuscript (Supplementary Data). Normalized data, counts/segments for all ROIs and genes, for FinXX are given in Supplementary Data 1, and data for Mayo Clinic TNBC TMA are given in Supplementary Data 7. Tumor-average data for FinXX samples are given in Supplementary Data 2. The remaining data are available within the Article or Supplementary Information.

## Code availability

The computer code used in these analyses is available at https://github.com/mayoclinicmayaohua/Finxx.

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

## Acknowledgements

The analyses described above were supported by grants from the Breast Cancer Research Foundation (BCRF 21-161) and the Florida Department of Health Bankhead Coley program (22B08) to E.A.T., the Susan G. Komen Foundation (SAC190091) to E.A.P., and the National Institutes of Health, USA, National Cancer Institute (2P50CA116201) to M.P.G. Additional support was provided by the Mayo Clinic Comprehensive Cancer Center grant P30CA15083 from the National Cancer Institute, National Institutes of Health, USA. R.L.F. was supported by CTSA Grant KL2 TR002379 from the National Center of Advancing Translational Science, USA.

## Author contributions

J.M.C., S.C., and E.A.T wrote the original manuscript, designed experiments, and performed all data analyses. D.A.H., S.W., H.A.B., and D.H. designed experiments and participated in computational analyses. J.M.K., J.S., and Y.L. performed experiments and were responsible for data management. Y.M., X.W., D.Z., D.W.H., and K.S.T. performed computational and statistical analyses. M.P.G., E.A.P., and E.A.T. supervised the work. H.J., H.L., R.A.F., J.C.B., M.C.L., J.N.I., K.R.K., F.J.C., and K.L.K. contributed to resources, and all authors contributed to manuscript writing/editing.

## Competing interests

D.A.H., S.E.W., H.A.B., and D.H. were employees of NanoString Technologies, Inc., at the time these experiments were carried out. J.M.C. declares consulting fees/advisory board fees (outside the scope of this work) from AstraZeneca, Agilent, Merck, and Roche. H.A.B. declares that the findings and conclusions contained within this manuscript are those of the authors and do not necessarily reflect the official positions or policies of her current employer, the Bill & Melinda Gates Foundation. M.C.L. is currently an employee of Natera. At the time these experiments were ongoing, M.C.L. received research support (but no personal compensation) from Eisai, Exact Sciences, Genentech, Genomic Health, GRAIL, Menarini Silicon Biosystems, Merck, Novartis, Seattle Genetics, and Tesaro. M.C.L. also served in an advisory capacity (no personal compensation) to Adela, AstraZeneca, Celgene, Roche/Genentech, Genomic Health, GRAIL, Ionis, Merck, Pfizer, Seattle Genetics, and Syndax. F.J.C. reports research support from GRAIL, consulting support from AstraZeneca, and speaker's fees from Natera and Ambry Genetics. R.L.F. reports consulting services for Gilead Sciences, AstraZeneca, and Lyell Immunopharma. Honoraria for services have been paid to Mayo Clinic for research activity, but no personal payments have been received by F.J.C. J.C.B. reports the payment to Mayo Clinic for consultation with Eli Lilly, SymBioSis, and Cairns Surgical, as well as speaker's fees from PER and PeerView and royalties for writing for UpToDate. H.J. reports personal compensation from Deciphera Pharmaceuticals, Maud Kulstila Foundation, Neutron Therapeutics, Orion Pharma, and Satar Therapeutics, as well as stocks/shares in Orion Pharma. M.P.G. reports personal fees for CME activities from Research to Practice, Clinical Education Alliance, Medscape, and MJH Life Sciences; personal fees for serving as a panelist for a panel discussion from Total Health Conferencing and personal fees for serving as a moderator for Curio Science; consulting fees to Mayo Clinic from ARC Therapeutics, AstraZeneca, Biotheranostics, Blueprint Medicines, Lilly, RNA Diagnostics, Sanofi Genzyme, and Seattle Genetics; and grant funding to Mayo Clinic from Lilly, Pfizer, and Sermonix. S.C. received research support, paid to Mayo Clinic, from Merck & Co., Pfizer, Salix Pharmaceuticals, and Rebiotix, Inc. S.C. received consulting fees, paid to Mayo Clinic, from AstraZeneca, Daiichi Sankyo, Immunomedics, Biotheranostics, Novartis, Athenex, Syndax, Puma Biotechnology, Eisai, and Seagen. K.L.K. reports consulting fees from Leidos, Antigen Express, and Affyimmune, with grant and other funding to Mayo Clinic from Marker Therapeutics, Macrogenics, Bolt Therapeutics, and Tallac, as well as royalties from Marker Therapeutics and stocks from Kiromic, Inc. The remaining authors declare no other competing interests.

## Additional information

[1]Department of Laboratory Medicine and Pathology, University of Alberta, Edmonton, Canada. [2]Department of Medicine, Division of Hematology and Oncology, Mayo Clinic, Jacksonville, FL, USA. [3]Ultima Genomics, Newark, CA, USA. [4]Department of Health Science Research, Division of Biomedical Statistics and Informatics, Mayo Clinic, Jacksonville, FL, USA. [5]Department of Health Science Research, Division of Biomedical Statistics and Informatics, Mayo Clinic, Rochester, MN, USA. [6]Department of Cancer Biology, Mayo Clinic, Jacksonville, FL, USA. [7]Bill and Melinda Gates Foundation, Seattle, WA, USA. [8]Kite Pharma, Santa Monica, CA, USA. [9]NanoString Technologies, Inc, Seattle, WA, USA. [10]Department of Oncology, Helsinki University Hospital and University of Helsinki, Helsinki, Finland. [11]Department of Oncology, University of Uppsala, Uppsala, Sweden. [12]Department of Oncology, Division of Medical Oncology, Mayo Clinic, Rochester, MN, USA. [13]Department of Surgery, Mayo Clinic, Rochester, MN, USA. [14]Natera, Inc, San Carlos, CA, USA. [15]Department of Laboratory Medicine and Pathology, Mayo Clinic, Rochester, MN, USA. [16]Department of Immunology, Mayo Clinic, Jacksonville, FL, USA. [17]These authors contributed equally: Jodi M. Carter, Saranya Chumsri, Douglas A. Hinerfeld. [18]These authors jointly supervised this work: Matthew P. Goetz, Edith A. Perez, E. Aubrey Thompson. ✉e-mail: thompson.aubrey@mayo.edu

