## [Peer Review File · Nature Communications]

Distinct spatial immune microlandscapes are independently associated with outcomes in triple negative breast cancerREVIEWER COMMENTS

Reviewer #1 (Remarks to the Author): with expertise in breast cancer

The authors analyzed the relationship between the spatial distribution of immune cell phenotypes in triple-negative breast cancer (TNBC) and outcomes (RFS or iDFS) in FinXX and a Mayo Clinic cohort. The spatial distribution of markers for immune cell phenotypes was determined using NanoString GeoMX Digital Spatial Profiling (DSP). The main finding is that intraepithelial localization of markers for antigen-presenting cells appears to be prognostic for TNBC outcomes, independent of other variables including stromal TILs.

General Comments:

1. It remains to be seen if extremely detailed knowledge of the spatial relationships of immune cells to tumor cells in TNBC will provide significantly more/better prognostic or predictive information than visual assessment of TILs. Extremely detailed knowledge of the spatial relationships of immune cells to tumor cells in TNBC could provide biological insights that, in principle, could lead to therapeutic insights, but this also has not been proven and can't really be addressed by the data presented in this paper. Furthermore, HLA-DR may be expressed by tumor cells and immune cells, so it's possible that one of the central findings in the paper has nothing to do with detailed spatial relationships between immune cells and tumor cells. The HLA-DR data may simply reflect tumor expression of a marker that showed a strong association with outcomes in the author's dataset.
2. The comparison of the data from intraepithelial segments to CD45+ stromal cells (as a surrogate for stromal TILs) in FinXX or sTILs in the Mayo cohort needs to be clarified. Stromal TILs are usually assessed on a full-face section of the tumor, if available. The intraepithelial segments are derived from part of a 300-micron or 600-micron ROI, likely limiting a direct comparison. If the point of the paper is that intraepithelial localization of immune cell markers is important for TNBC outcomes, why not also assess intraepithelial TILs (iTILs) (per the guidelines from the International Immuno-Oncology Biomarker Working Group on Breast Cancer) in FinXX or the Mayo cohort? Are the sTILs data in the Mayo cohort based on the TMA or the whole slide/block from which the area of the TMA core was obtained? Why were sTILs on FinXX unavailable? Are the H&E slides from the 44 cases from FinXX unavailable?

Detailed Comments:

1. Introduction, 2nd paragraph: "...stromal TIL scoring does not provide spatial insight..." By definition, doesn't the evaluation of stromal TILs versus intraepithelial TILs provide at least some spatial insight?
2. Introduction, 2nd paragraph: "The importance of spatial context in tumoral immune microenvironments (TIME) is underscored by recent data describing distinct immune phenotypes in TNBC,..." There are recent peer-reviewed publications on TNBC that should be cited instead of or in addition to the 2017 review and papers on melanoma and lung cancer.

Mi H, Gong C, Sulam J, Fertig EJ, Szalay AS, Jaffee EM, Stearns V, Emens LA, Cimino-Mathews AM, Popel AS. Digital Pathology Analysis Quantifies Spatial Heterogeneity of CD3, CD4, CD8, CD20, and FoxP3 Immune Markers in Triple- Negative Breast Cancer. *Front Physiol.* 2020 Oct 19;11:583333. doi: 10.3389/fphys.2020.583333. PMID: 33192595; PMCID: PMC7604437.

Walens A, Olsson LT, Gao X, Hamilton AM, Kirk EL, Cohen SM, Midkiff BR, Xia Y, Sherman ME, Nikolaishvili-Feinberg N, Serody JS, Hoadley KA, Troester MA, Calhoun BC. Protein-based immune profiles of basal-like vs. luminal breast cancers. *Lab Invest.* 2021 Jun;101(6):785-793. doi: 10.1038/s41374-020-00506-0. Epub 2021 Feb 23. PMID: 33623115; PMCID: PMC8140991.

3. Methods, Patient Samples, 2nd paragraph: How was size of the subset (n=44) from FinXX determined?
4. Methods, Digital Spatial Profiling: Was the failure of QC in all TumorEnR stroma segments in FinXX related to the size of the ROI – since the CK segmentation was the same for FinXX and the Mayo cohort and the TumorEnR stroma segments from the Mayo Cohort presumably passed QC? How

does the deletion of the TumorEnR stroma segments from FinXX limit what we can conclude about the importance of intraepithelial versus stromal localization of immune cell markers?

5. Results: “Conventional histopathological sTIL scores were not available for FinXX samples. We used quantitative image analysis of the GeoMx™ digital IF composite images to quantify stromal CD45+ cells and to assort tumors into tertiles by stromal CD45+ cell density per mm².” The authors should review the H&Es for FinXX and include a conventional visual assessment of stromal and/or intraepithelial TILs.

6. Discussion: “Although stromal TIL scores have established prognostic value, they serve as a surrogate of a complex interplay of immune cell populations, cannot provide biological or clinically relevant insights into the immune activation status of the stromal immune milieu, and do not directly elucidate the immune landscape of the intraepithelial compartment.” TILs have been shown to have prognostic and predictive value in thousands of patients from multiple clinical trials. Although we usually assess stromal TILs, intraepithelial TILs (iTILs) can be assessed per guidelines provided by the International Immuno-Oncology Biomarker Working Group on Breast Cancer. By definition TILs include all mononuclear cells, so they would include all of the cell types in the authors’ analysis. Why didn’t the authors assess iTILs in FinXX and the Mayo Cohort? Or sTILs in FinXX?

Reviewer #3 (Remarks to the Author): with expertise in digital spatial profiling and breast cancer

Carter et al use high-plex digital spatial profiling (DSP, NanoString GeoMx) to detect differences in levels of expression of tumor related immune proteins between intraepithelial and stromal segments of early-stage Triple Negative Breast Cancer (TNBC) samples. The researchers use two independent TNBC cohorts to explore protein expression differences and associations with clinical outcome. 44 pre-treated whole tissue samples from patients enrolled in the FinXX trial and 275 tumor samples from a TNBC cohort-derived TMA were tested the GeoMx digital spatial profiling technology. They found that immune cells in close contact with tumor cells (intra-epithelial lymphocytes or iTIL) were better predictors than immune cells in the stroma (stromal lymphocytes or sTIL).

Overall, the study highlights the importance of spatial distribution of specific subtypes of immune infiltrates in predicting clinical outcome. This is a novel and important finding. While iTIL and sTIL are not novel, pathologists consistently fail to accurately assess iTIL and thus have relied on sTIL. Here, using molecular and spatial tools the authors show that iTIL are more important than sTIL.

There are a number of limitations that are listed below, and some suggested changes that would improve the presentation of the work and make the key conclusions more clear:

1. The researchers compare tumor immune protein microenvironments (TIME) in stromal and intraepithelial segments. Suppl. figure 1 is the only figure showing the two segments. Since the spatial context information plays a main role in the study, a magnified representative image of the two segments would be helpful to visualize the two segments. In addition, a representative figure of the TMA ROI segmentation would be useful.

2. The nomenclature is confusing. Although two molecular compartments are described for each ROI, one is CK positive and the other CK negative. CK positive compartment consists mostly of tumor cells yet it is named intraepithelial segment (which is the source of the confusion). While I understand that it represents the tumor cells and closely adjacent stroma, calling it the intraepithelial segment does not convey tumor and iTIL to me. TumorEnR is also not well defined and seems to introduce subjectivity. Are EnRs subjectively defined? Please explain and justify.

If the researchers had examined compartments based on CD45 positivity cells in the tumor mask and in stromal mask then the division of this cell population into intraepithelial and stromal segments would make sense to me. There is precedence in the nomenclature of AQUA-based analysis that might be easier to define and understand. A “mask” is the CK+ region with adjacent stroma limited by the resolution of the technology and may include dilation or hole filling, where a “compartment” is strictly the pixels defined by the molecular interaction. CD45 cells may be present in a CK+ mask but should not be present in a CK+ compartment and historically, segments are defined by feature extraction, so I am not sure what a CK+ segment means.

The best nomenclature is a subjective thing since no standards exist. The term segmentation or segment is commonly used to infer feature extraction which is not done here. It may be best to use the term molecularly-defined compartment or something similar to prevent confusion related to segments defined by feature extraction software.

3. Page 6, line 14-15: Tumor samples...therapy. From 605 patients 275 patient samples were examined and 2/3 of the 275 patients received adjuvant chemotherapy whereas 1/3 did not receive systemic therapy. Confusing or error? Please clarify.

4. Page 7, line 20-23: One 600-micron...area. It is not clear what the authors mean here. Did they select more than 1 ROI/ core? Was the TMA 2-fold redundant, meaning for each patient you had 2 cores? The authors state that ROIs were not targeted based on stromal enrichment; does that mean that the CK negative segment could consist of both CD68 positive and CD45 positive cells? It would help if this is rephrased and accompanied by illustrative schematics or images.

5. FinXX trial samples have survival data available (RFS). For segment level analysis the researchers choose to calculate odds ratios treating RFS as a binomial categorical variable (Recurrence YES or NO). Please explain why you didn't choose analysis based on Cox-proportional HR to account for the different time points of each patient's recurrence for the FinXX samples as you did with the TMA samples.

6. Tertiles were used as a threshold for high vs low expression of the target proteins. How was this cut-point selected (rather than median or quartiles or ???)?

7. In the text, page 13, line 17, based on Figure 1, the authors show the distribution of 9 proteins (PD-L2, PD-L1, GZMB, CD20, ICOS, IDO1, HLA-DR, CD56, CD40) which were associated with RFS, then they calculate OR for each of these proteins. They also calculate OR for CD11c. Why?

8. Page 15, bottom paragraph states that sTIL was not available for FinXX. It would be more compelling to have pathologist scores that quantification with the IF composite images. This is a little confusing since the paper, to this point, is showing iTIL are better than sTIL, then in Supp figure 3 sTIL are shown to be prognostic.

9. Figure 1: Please correct the figure legend. Tumors that did not recur are shown in light grey not blue.

Reviewer #4 (Remarks to the Author): with expertise in spatial profiling and tumor immune microenvironment

Carter et al present a proteomic analysis of TNBC microenvironment. They use DSP to catalogue the expression of a number of different proteins from different regions of tumor samples. They construct masks based on keratin, CD45, and CD68 to identify different compartments of each tumor, and quantitate the expression of these different proteins in each compartment. They find different proteins which are upregulated in each of these compartments, some of which are correlated with patient outcome. They then replicate this analysis in a separate cohort of TNBC patients. Finally, they look at the correlation between different proteins in their panel.

Overall, this study is technically sound, and the use of two orthogonal cohorts is a major strength. However, in its current form, it is difficult to appreciate what biological insight is presented by the authors. The authors identify expression of antigen-presentation as a positive prognostic factor, but this is a well-established trend across many different tumor types, and has been noted specifically for TNBC in multiple previous publications (for example, see PMIDs 28817603, 26980599, 30193111). Forero et al. is actually referenced in the discussion, but it was not noted that this previous study was

similar in size to the current one and included bulk RNA seq with confirmatory laser capture microdissection and single plex IHC, the latter being something that is notably missing here. The compartment-specific nature of this effect is interesting, but given the limitations of the DSP antibody platform, the authors are not able to follow-up further on this trend. The correlation analyses are likewise not very informative. Overall, this is a well-designed study with rigorous statistical analyses, but the novelty of the findings is limited.

Major Concerns:

1. One of the major findings is that proteins associated with antigen presentation are enriched in the intraepithelial segments. While HLA-DR can be expressed in both tumor cells or APCs, the authors are not able to identify the cell population responsible for this increased expression. The authors cite one of the prior studies demonstrating that intraepithelial HLA-DR was associated with better clinical outcome, so it is unclear what additional insight this study presents. Additionally, multiple avenues exist to probe this directly (i.e. IF with HLA-DR, CD45, and PanCK) that would not require the expensive and cumbersome techniques employed here.
2. While the authors perform sound statistical analysis, the motivation behind performing these analyses is unclear. For example, the correlation analysis shown in Table 3 largely recapitulates known biology (i.e. it is not informative that T cell and APC markers are correlated)
3. The analysis in the "Spatial context determines immune protein association with outcome" section is quite repetitive. The authors first identify the proteins which are associated with RFS (Table 1). The authors then plot the expression of these proteins, and find differences in expression between patients with good outcome and patients with poor outcome (Figure 1). However, these proteins were selected because they were associated with outcome, so the purpose of this figure is unclear. Next, the authors take these same set of proteins, which we know are associated with survival, and demonstrate that there are significant ORs when binning patients based off their expression level (Table 2). Finally, using these same proteins, the authors construct Kaplan-Meier curves, and find differences in outcome (Figure 2), which is once again not surprising

Minor Concerns:

1. The LFC values are fairly modest, even for markers which are explicitly selected for ROI creation. Is there a reason that a greater difference in expression was not observed?
2. The authors use fold change throughout the paper, which can be unstable for low-expressing proteins. A high fold change could be driven by low expression, which may lead to misleading conclusions. In particular, the authors state that PD-L2 was the only protein associated with durable recurrence free survival in any stromal segment class, and also state that PD-L2 was expressed at low levels. Along these lines, we have found PD-L2 to in the vast majority of cases have weak and non-specific staining with all commercial clones we've tested, making this finding further suspect without confirmatory single-plex imaging. This significant fold change could be due to low expression. It would be helpful for the authors to include the baseline expression levels as well, or a violin plot.
3. It would be helpful to include significance values in Figure 1.
4. The authors state that the limited correlation between total CD45 levels and the other immune proteins they identified means that there is independent prognostic information in these proteins beyond sTIL levels (top of page 16). However, this claim requires a multivariate model showing independent significance to verify that this is the case
5. It is surprising that CD20 does not correlate with CD45, since presumably cells that are expressing CD20 (B cells) are also expressing CD45. Some more explanation about these findings would be helpful. These results could potentially be biased by non-specific antibody binding.
6. Similarly, B cells are antigen presenting cells and express HLA-DR, and it is a little surprising CD20 does not correlate with HLA-DR. It seems in Figure 3a that they are more coupled in the stroma. It would be helpful to comment on these findings in more detail.
7. In the differential expression plots (Supplementary Figure 2 and 5), it would be helpful to sort by average fold change to be able to see patterns more clearly.
8. The authors state that they observed similar LFCs in the Mayo and FinXX cohort, but this is difficult to evaluate since they are presented in different supplementary figures. A single figure which has one

cohort on the X axis and the other cohort on the Y axis would be an easier way to visually assess the similarities between the two datasets

9. While the authors mention in the discussion that the GeoMx platform cannot provide single-cell level data and therefore cannot be used to assess single-cell level expression, it is possible to use the fluorescent images used to guide region selection to get an approximation for the number of CD45+ and cytokeratin+ cells (i.e. immune and tumor cells). These counts could be informative to include to assess whether there is more immune infiltrate, or a similar number of cells with cells expressing a higher level of protein.

10. The authors have not made their code or data available. This should be remedied prior to publication.

REVIEWER COMMENTS

Reviewer #1 (Remarks to the Author): with expertise in breast cancer

Comment to all reviewers:

This manuscript has undergone a major revision. Revisions were undertaken to address, insofar as possible, all specific questions raised in the initial review. Most of the repetitive data to which some reviewers objected has been removed, as well as other observations that were confusing or poorly described. We have carried out several major new analyses in an attempt to provide a more acute focus on those observations that we perceive to be of particular novelty and/or significance with respect to the immunological landscape of triple negative breast cancer as well as the relationship between these immunological features and clinical phenotype (outcome) in early stage TNBC. We submit, without fear of contradiction, that this is the largest and most comprehensive analysis of the spatial immune landscape in triple negative breast cancer that has ever been reported. We have taken advantage of the quantitative (digital) nature of our data to develop novel immune scores that consolidate the abundance of key antigen presenting and T cell activation features to elucidate the relationship between these processes and clinical outcome. We have shown that the relevant features associated with outcome are largely confined to the intraepithelial space within tumors, in which immune cells are in direct communication with tumor cells. We have shown that these key features are associated with outcome in a manner that is not dependent on the abundance of stromal TILs. Our studies show, for the first time, that the prognostic significance of PD-L1 expression depends upon the context of intraepithelial antigen-presenting and T cell activation status. This observation, alone, has tremendous potential clinical impact. We have shown that IDO1, unlike PD-L1, is associated with outcome even in tumors with low intraepithelial antigen-presenting activity. However, the prognostic significance of IDO1 is dependent upon association with intraepithelial T cell activation status. In short, we feel that this major revision has materially increased the significance, novelty, and impact of our comprehensive analyses of two independent cohorts of early-stage triple negative breast cancer.

The authors analyzed the relationship between the spatial distribution of immune cell phenotypes in triple-negative breast cancer (TNBC) and outcomes (RFS or iDFS) in FinXX and a Mayo Clinic cohort. The spatial distribution of markers for immune cell phenotypes was determined using NanoString GeoMX Digital Spatial Profiling (DSP). The main finding is that intraepithelial localization of markers for antigen-presenting cells appears to be prognostic for TNBC outcomes, independent of other variables including stromal TILs.

General Comments:

1. It remains to be seen if extremely detailed knowledge of the spatial relationships of immune cells to tumor cells in TNBC will provide significantly more/better prognostic or predictive information than visual assessment of TILs. Extremely detailed knowledge of the spatial relationships of immune cells to tumor cells in TNBC could provide biological insights that, in principle, could lead to therapeutic insights,

but this also has not been proven and can't really be addressed by the data presented in this paper. Furthermore, HLA-DR may be expressed by tumor cells and immune cells, so it's possible that one of the central findings in the paper has nothing to do with detailed spatial relationships between immune cells and tumor cells. The HLA-DR data may simply reflect tumor expression of a marker that showed a strong association with outcomes in the author's dataset.

RESPONSE: We agree with the reviewer. We have not proved that the detailed knowledge of the spatial relationship between immune cells and tumor cells will lead to actionable therapeutic insights. Such a proof of concept would require a prospective analysis (clinical trial) or a retrospective analysis of the predictive potential of our findings within the context of therapeutic response to immune checkpoint inhibition in TNBC. Either of these approaches represents a major undertaking well beyond the scope of this initial analysis of over 300 TNBC samples. Be that as it may, we remain convinced that effective therapeutic strategies must be based on a solid understanding of the immune biology of TNBC, and we submit that our analyses provide a depth of understanding of the immune architecture of TNBC that has not been available heretofore. We are motivated by the conviction that the development of novel therapeutic strategies must rest on a fundamental understanding of the immune features that we have described in our studies.

2. The comparison of the data from intraepithelial segments to CD45+ stromal cells (as a surrogate for stromal TILs) in FinXX or sTILs in the Mayo cohort needs to be clarified. Stromal TILs are usually assessed on a full-face section of the tumor, if available. The intraepithelial segments are derived from part of a 300-micron or 600-micron ROI, likely limiting a direct comparison. If the point of the paper is that intraepithelial localization of immune cell markers is important for TNBC outcomes, why not also assess intraepithelial TILs (iTILs) (per the guidelines from the International Immuno-Oncology Biomarker Working Group on Breast Cancer) in FinXX or the Mayo cohort?

RESPONSE: As the reviewer knows, and as stated by Reviewer 3, evaluation of intraepithelial TILs has proved to be highly problematic, both in terms of intra-operator and inter-operator reproducibility. The clinical significance of iTILs remains to be established. Conversely, the quantitative digital nature of the DSP data provides a tool for a more precise and reliable evaluation of the immune landscape within tumors nests (the so-called intraepithelial compartment). We have shown that we can reliably and reproducibly interrogate the intraepithelial domains and use the data from such analyses to begin to build models that inform prognosis in TNBC and elucidate the extent to which the several immune features (e.g., antigen-presentation, T cell activation, and PD-L1 abundance) interact so as to influence durable recurrence free survival. Given the well-known technical issues associated with quantification of iTILs, it seems unlikely that such models will be built on iTIL data. Moreover, it should be emphasized that analysis of TILs, in general, focuses only on easily identified lymphoid cells; whereas DSP analysis, as shown, facilitates analysis of the immune landscape associated with other cell types, including macrophages and antigen-presenting cells.

Are the sTILs data in the Mayo cohort based on the TMA or the whole slide/block from which the area of the TMA core was obtained? Why were sTILs on FinXX unavailable? Are the H&E slides from the 44 cases from FinXX unavailable?

RESPONSE: TILs for both the Mayo cohort and now the FinXX samples (new data in revision) were obtained from full face sections. For the revision, we obtained H&E-stained slides from the 44 FinXX samples that we analyzed. These were used to measure TILs, by Dr. Carter, a breast pathologist who heads the Mayo Clinic breast cancer biomarker program. Our analyses show that intraepithelial antigen-presenting activity has prognostic significance even when TILs are included as a covariable. Moreover, our data show that antigen-presenting activity is associated with improved outcome even when quantified in tumor segments that contain few or no adjacent stromal lymphoid cells. We submit that our findings emphasize the importance of evaluating the significance of those immune features that are associated with the intraepithelial compartment of TNBC at a level of detail that goes well beyond iTILs.

Detailed Comments:

1. Introduction, 2nd paragraph: "...stromal TIL scoring does not provide spatial insight..." By definition, doesn't the evaluation of stromal TILs versus intraepithelial TILs provide at least some spatial insight?

RESPONSE: The reviewer is correct. Obviously, analysis of stromal TILs provides a degree of spatial insight. The sentence in question has been modified to focus on the fact that histological evaluation of TILs provides only limited insight into the cellular diversity or activity of a subset of immune cells (primarily T-cells), whereas DSP captures data related to the amount and/or activity of other key immune cell types that cannot be reliably inferred from conventional analysis of TILs.

It may be worth emphasizing that we are not challenging the clinical utility of sTILs in TNBC. Our focus has been to determine to what extent we can map the immune landscape derived from immune cells that are in direct contact with tumor cells, within the intraepithelial segments, and use this information to better understand the relationship between tumor cell-immune cell interaction and clinical outcome.

2. Introduction, 2nd paragraph: "The importance of spatial context in tumoral immune microenvironments (TIME) is underscored by recent data describing distinct immune phenotypes in TNBC,..." There are recent peer-reviewed publications on TNBC that should be cited instead of or in addition to the 2017 review and papers on melanoma and lung cancer.

Mi H, Gong C, Sulam J, Fertig EJ, Szalay AS, Jaffee EM, Stearns V, Emens LA, Cimino-Mathews AM, Popel AS. Digital Pathology Analysis Quantifies Spatial Heterogeneity of CD3, CD4, CD8, CD20, and FoxP3 Immune Markers in Triple- Negative Breast Cancer. *Front Physiol.* 2020 Oct 19;11:583333. doi: 10.3389/fphys.2020.583333. PMID: 33192595; PMCID: PMC7604437.

Walens A, Olsson LT, Gao X, Hamilton AM, Kirk EL, Cohen SM, Midkiff BR, Xia Y, Sherman ME, Nikolaishvili-Feinberg N, Serody JS, Hoadley KA, Troester MA, Calhoun BC. Protein-based immune

profiles of basal-like vs. luminal breast cancers. Lab Invest. 2021 Jun;101(6):785-793. doi: 10.1038/s41374-020-00506-0. Epub 2021 Feb 23. PMID: 33623115; PMCID: PMC8140991.

RESPONSE: We thank the reviewer for pointing out this oversight. These references have been included in the revised manuscript.

3. Methods, Patient Samples, 2nd paragraph: How was size of the subset (n=44) from FinXX determined?

RESPONSE: Among the 110 samples for which we had good quality slides, there were 24 samples that had a recurrence event. We selected 11 samples that recurred from each arm (5FU and capecitabine). We then selected 11 matched controls (no recurrence) from each arm, for a total of 44 samples. This information has been added to the Materials and Methods section.

4. Methods, Digital Spatial Profiling: Was the failure of QC in all TumorEnR stroma segments in FinXX related to the size of the ROI – since the CK segmentation was the same for FinXX and the Mayo cohort and the TumorEnR stroma segments from the Mayo Cohort presumably passed QC?

RESPONSE: The failure of QC in the TumorEnR stroma segments was not related to ROI size. All ROIs were of the same size. Rather, we purposefully selected TumorEnR ROIs that had few or no nucleated cells in the adjacent stroma—they are dense tumor nests with little or no stroma. Based on SYTO13 (nuclear) staining, these stromal TumorEnR segments contained fewer than 30 nuclei and were therefore filtered from subsequent analysis according to NanoString DSP best practice guidelines. The Methods section has been modified to clarify this point.

How does the deletion of the TumorEnR stroma segments from FinXX limit what we can conclude about the importance of intraepithelial versus stromal localization of immune cell markers?

RESPONSE: It would be difficult to know how to evaluate the immune landscape of stromal segments that contain no immune or other type cells. Apropos of our analysis of the intraepithelial segments, we note that by and large there were fewer immune cells in the TumorEnR intraepithelial segments. Nevertheless, the relationship between outcome and intraepithelial protein abundance was largely the same, irrespective of the immune cell composition of the adjacent stroma. We submit that this is a significant finding. It is not intuitively obvious that the relationship between antigen-presenting activity and outcome would be more or less independent of the immune cell composition of the adjacent stroma.

5. Results: “Conventional histopathological sTIL scores were not available for FinXX samples. We used quantitative image analysis of the GeoMxTM digital IF composite images to quantify stromal CD45+ cells

and to assort tumors into tertiles by stromal CD45+ cell density per mm².” The authors should review the H&Es for FinXX and include a conventional visual assessment of stromal and/or intraepithelial TILs.

RESPONSE: For the revised manuscript, we obtained additional slides from FinXX. As requested, we evaluated stromal TILs in these samples and used the TIL data in multivariable analysis. It may be worth noting that the small size of the FinXX cohort (only 22 events) precluded a more comprehensive multivariable analysis. However, such an analysis was carried out in the much larger Mayo Clinic TNBC cohort.

6. Discussion: “Although stromal TIL scores have established prognostic value, they serve as a surrogate of a complex interplay of immune cell populations, cannot provide biological or clinically relevant insights into the immune activation status of the stromal immune milieu, and do not directly elucidate the immune landscape of the intraepithelial compartment.” TILs have been shown to have prognostic and predictive value in thousands of patients from multiple clinical trials. Although we usually assess stromal TILs, intraepithelial TILs (iTILs) can be assessed per guidelines provided by the International Immuno-Oncology Biomarker Working Group on Breast Cancer. By definition TILs include all mononuclear cells, so they would include all of the cell types in the authors’ analysis. Why didn’t the authors assess iTILs in FinXX and the Mayo Cohort? Or sTILs in FinXX?

RESPONSE : As mentioned, we have now obtained H&E-stained slides from the FinXX cohort and scored sTILs (see comments above to reviewer #1). With regard to iTILs, iTIL scoring has low interobserver reproducibility and scoring iTILs is not currently endorsed by the International Immune Biomarker Working Group for prognostication in TNBC (as noted by reviewer 3 below). For these reasons, we did not score iTILs. Rather, we used sTIL data from both the FinXX and Mayo TNBC cohorts to compare to the intraepithelial quantitative immune biomarker data to address whether sTILs reflect the intraepithelial immune activity, and secondly as a variable in multivariate survival models.

Reviewer #3 (Remarks to the Author): with expertise in digital spatial profiling and breast cancer

Comment to all reviewers:

This manuscript has undergone a major revision. Revisions were undertaken to address, insofar as possible, all specific questions raised in the initial review. Most of the repetitive data to which some reviewers objected has been removed, as well as other observations that were confusing or poorly described. We have carried out several major new analyses in an attempt to provide a more acute focus on those observations that we perceive to be of particular novelty and/or significance with respect to the immunological landscape of triple negative breast cancer as well as the relationship between these immunological features and clinical phenotype (outcome) in early stage TNBC. We submit, without fear of contradiction, that this is the largest and most comprehensive analysis of the spatial immune landscape in triple negative breast

cancer that has ever been reported. We have taken advantage of the quantitative (digital) nature of our data to develop novel immune scores that consolidate the abundance of key antigen presenting and T cell activation features to elucidate the relationship between these processes and clinical outcome. We have shown that the relevant features associated with outcome are largely confined to the intraepithelial space within tumors, in which immune cells are in direct communication with tumor cells. We have shown that these key features are associated with outcome in a manner that is not dependent on the abundance of tumor-infiltrating lymphocytes. Our studies show, for the first time, that the prognostic significance of PD-L1 expression is dependent upon the context of intraepithelial antigen-presenting and T cell activation status. This observation, alone, has tremendous potential clinical impact. We have shown that IDO1, unlike PD-L1, is associated with outcome even in tumors with low intraepithelial antigen-presenting activity. However, the prognostic significance of IDO1 is dependent upon association with intraepithelial T cell activation status. In short, we feel that this major revision has materially increased the significance, novelty, and impact of our comprehensive analyses of two independent cohorts of early stage triple negative breast cancer.

Carter et al use high-plex digital spatial profiling (DSP, NanoString GeoMx) to detect differences in levels of expression of tumor related immune proteins between intraepithelial and stromal segments of early-stage Triple Negative Breast Cancer (TNBC) samples. The researchers use two independent TNBC cohorts to explore protein expression differences and associations with clinical outcome. 44 pre-treated whole tissue samples from patients enrolled in the FinXX trial and 275 tumor samples from a TNBC cohort-derived TMA were tested the GeoMx digital spatial profiling technology. They found that immune cells in close contact with tumor cells (intra-epithelial lymphocytes or iTIL) were better predictors than immune cells in the stroma (stromal lymphocytes or sTIL).

Overall, the study highlights the importance of spatial distribution of specific subtypes of immune infiltrates in predicting clinical outcome. This is a novel and important finding. While iTIL and sTIL are not novel, pathologists consistently fail to accurately assess iTIL and thus have relied on sTIL. Here, using molecular and spatial tools the authors show that iTIL are more important than sTIL.

There are a number of limitations that are listed below, and some suggested changes that would improve the presentation of the work and make the key conclusions more clear:

1. The researchers compare tumor immune protein microenvironments (TIME) in stromal and intraepithelial segments. Suppl. figure 1 is the only figure showing the two segments. Since the spatial context information plays a main role in the study, a magnified representative image of the two segments would be helpful to visualize the two segments. In addition, a representative figure of the TMA ROI segmentation would be useful.

RESPONSE: High magnification images have been added to Supplementary Figure 1.

2. The nomenclature is confusing. Although two molecular compartments are described for each ROI, one is CK positive and the other CK negative. CK positive compartment consists mostly of tumor cells yet it is named intraepithelial segment (which is the source of the confusion). While I understand that it represents the tumor cells and closely adjacent stroma, calling it the intraepithelial segment does not convey tumor and iTIL to me. TumorEnR is also not well defined and seems to introduce subjectivity. Are EnRs subjectively defined? Please explain and justify. If the researchers had examined compartments based on CD45 positivity cells in the tumor mask and in stromal mask then the division of this cell population into intraepithelial and stromal segments would make sense to me. There is precedence in the nomenclature of AQUA-based analysis that might be easier to define and understand. A “mask” is the CK+ region with adjacent stroma limited by the resolution of the technology and may include dilation or hole filling, where a “compartment” is strictly the pixels defined by the molecular interaction. CD45 cells may be present in a CK+ mask but should not be present in a CK+ compartment and historically, segments are defined by feature extraction, so I am not sure what a CK+ segment means. The best nomenclature is a subjective thing since no standards exist. The term segmentation or segment is commonly used to infer feature extraction which is not done here. It may be best to use the term molecularly-defined compartment or something similar to prevent confusion related to segments defined by feature extraction software.

RESPONSE: The new Supplementary Figure 1 shows magnified representations of the segmentation results from FinXX. We did not include similar figures from the TMA, since the segmentation results are identical to those from FinXX.

We apologize for the confusion concerning the cellular composition of the stroma and intraepithelial segments. The intraepithelial compartment does not represent tumor cells plus closely adjacent stroma. Rather, the intraepithelial compartment represents the “tumor cell nests”, which contain cytokeratin-positive tumor cells plus cytokeratin-negative tumor-cell associated immune cells. The text has been modified in an attempt to make this point more clearly.

Initially, we did not segment on CD45 because we did not wish to restrict our analysis to CD45-positive immune cells. Our objective was to quantify immune proteins within the stroma and the cytokeratin-positive tumor nests (the intraepithelial compartment defined by the CK+ area of illumination and consisting of tumor cells plus immune cells that had infiltrated into the tumor nests). We have expanded the description of the segmentation process in Materials and Methods to minimize the possibility of confusion concerning the cellular contents of the intraepithelial and stroma compartments.

The process of defining areas of illumination based on IF images is called “segmentation” by NanoString, and is described in these terms in the GeoMx user’s manual and the recently published “Best Practices” report (PMID: 34503266). We have adhered to this nomenclature so as to minimize any possibility of confusion by investigators who might seek to replicate our experiments while following the instructions in the user’s manual.

3. Page 6, line 14-15: Tumor samples...therapy. From 605 patients 275 patient samples were examined

and 2/3 of the 275 patients received adjuvant chemotherapy whereas 1/3 did not receive systemic therapy. Confusing or error? Please clarify.

RESPONSE: This confusing statement has been deleted.

4. Page 7, line 20-23: One 600-micron...area. It is not clear what the authors mean here. Did they select more than 1 ROI/ core?

RESPONSE: The text has been modified to read "A single 600 micron...". Only one ROI was selected for each core.

Was the TMA 2-fold redundant, meaning for each patient you had 2 cores?

RESPONSE: The TMA was constructed to contain 2 replicate cores from each patient. However, as described in the text, the dimensions of the TMA exceeded the field of view of the GeoMx camera. Some cores fell outside the field of view, so we did not capture 2 replicate cores for every sample. We averaged those samples for which we did capture 2 cores.

The authors state that ROIs were not targeted based on stromal enrichment; does that mean that the CK negative segment could consist of both CD68 positive and CD45 positive cells? It would help if this is rephrased and accompanied by illustrative schematics or images.

RESPONSE: We have rephrased this sentence to read "No attempt was made to select ROIs based on the immune cell composition of the adjacent stroma. Consequently, the TMA samples represent a random selection of CD45, CD68, or tumor-enriched ROIs."

5. FinXX trial samples have survival data available (RFS). For segment level analysis the researchers choose to calculate odds ratios treating RFS as a binomial categorical variable (Recurrence YES or NO). Please explain why you didn't choose analysis based on Cox-proportional HR to account for the different time points of each patient's recurrence for the FinXX samples as you did with the TMA samples.

RESPONSE: Both univariable and multivariable Cox HRs have been added to the text in Tables 2 (FinXX) and 3 (TNBC TMA). Five-year ORs have been re-calculated based on median abundance and are now included in Figure 3 and supplementary table 8.

6. Tertiles were used as a threshold for high vs low expression of the target proteins. How was this cut-point selected (rather than median or quartiles or ???)?

RESPONSE: Initially, the decision to use tertiles in the FinXX analysis was convenient to the sample size, 44 samples. Moreover, visual examination of the distribution of RFS YES/NO samples shown in Figure 1 suggested that about 1/3 of the samples exhibited high level expression associated with RFS. This suggestion was substantiated by the KM analyses shown in Figure 2.

7. In the text, page 13, line 17, based on Figure 1, the authors show the distribution of 9 proteins (PD-L2,

PD-L1, GZMB, CD20, ICOS, IDO1, HLA-DR, CD56, CD40) which were associated with RFS, then they calculate OR for each of these proteins. They also calculate OR for CD11c. Why?

RESPONSE: The decision to include CD11c in the FinXX analyses was made post hoc based on analysis of the Mayo Clinic samples, in which CD11c emerged as an important marker in multivariable analysis.

8. Page 15, bottom paragraph states that sTIL was not available for FinXX. It would be more compelling to have pathologist scores that quantification with the IF composite images. This is a little confusing since the paper, to this point, is showing iTIL are better than sTIL, then in Supp figure 3 sTIL are shown to be prognostic.

RESPONSE: We have now included new data on conventional histopathological evaluation of sTILs in the FinXX samples, including multivariable analysis showing that a number of intraepithelial features are prognostic, independent of sTILs.

To clarify, we did not state, nor did we intend to imply that iTILs were better than sTILs. We measured the spatial distribution of immune function proteins, not TILs, as a function of outcome. Some of these proteins are clearly markers of TILs (CD20, CD3, CD8 etc.), whereas others are markers of immune cell types that are not conventionally quantified in histopathological analysis of TILs (CD56, CD68, APC markers), and some are likely markers of immune cell activity rather than cell number (IDO1, GZMB, ICOS, etc.). The main point that we are striving to make is that analysis of the intraepithelial TIME reveals significant immune features that are not apparent in, or predicted by, analysis of the adjacent stroma.

Stated simply, we submit that there is more nuance to the immune landscape, particularly in the intraepithelial tumor immune micro-environment than sTIL scores can provide. This assertion may not seem to be particularly insightful; however, it is the case that most analysis of the immune architecture of TNBC has, to date, focused on the role of lymphocytes, primarily T-cells, in the stroma. Our data expand the analysis of the immune landscape of TNBC to include immune features other than those associated with lymphocytes in both stromal and intraepithelial microenvironments within the tumor.

9. Figure 1: Please correct the figure legend. Tumors that did not recur are shown in light grey not blue.

RESPONSE: Done.

Reviewer #4 (Remarks to the Author): with expertise in spatial profiling and tumor immune microenvironment

Comment to all reviewers:

This manuscript has undergone a major revision. Revisions were undertaken to address, insofar as possible, all specific questions raised in the initial review. Most of the repetitive data to which some reviewers objected has been removed, as well as other observations that were confusing or poorly described. We have carried out several major new analyses in an attempt to

provide a more acute focus on those observations that we perceive to be of particular novelty and/or significance with respect to the immunological landscape of triple negative breast cancer as well as the relationship between these immunological features and clinical phenotype (outcome) in early stage TNBC. We submit, without fear of contradiction, that this is the largest and most comprehensive analysis of the spatial immune landscape in triple negative breast cancer that has ever been reported. We have taken advantage of the quantitative (digital) nature of our data to develop novel immune scores that consolidate the abundance of key antigen presenting and T cell activation features to elucidate the relationship between these processes and clinical outcome. We have shown that the relevant features associated with outcome are largely confined to the intraepithelial space within tumors, in which immune cells are in direct communication with tumor cells. We have shown that these key features are associated with outcome in a manner that is not dependent on the abundance of tumor-infiltrating lymphocytes. Our studies show, for the first time, that the prognostic significance of PD-L1 expression depends upon the the context of intraepithelial antigen-presenting and T cell activation status. This observation, alone, has tremendous potential clinical impact. We have shown that IDO1, unlike PD-L1, is associated with outcome even in tumors with low intraepithelial antigen-presenting activity. However, the prognostic significance of IDO1 is dependent upon association with intraepithelial T cell activation status. In short, we feel that this major revision has materially increased the significance, novelty, and impact of our comprehensive analyses of two independent cohorts of early stage triple negative breast cancer.

Carter et al present a proteomic analysis of TNBC microenvironment. They use DSP to catalogue the expression of a number of different proteins from different regions of tumor samples. They construct masks based on keratin, CD45, and CD68 to identify different compartments of each tumor, and quantitate the expression of these different proteins in each compartment. They find different proteins which are upregulated in each of these compartments, some of which are correlated with patient outcome. They then replicate this analysis in a separate cohort of TNBC patients. Finally, they look at the correlation between different proteins in their panel.

Overall, this study is technically sound, and the use of two orthogonal cohorts is a major strength. However, in its current form, it is difficult to appreciate what biological insight is presented by the authors. The authors identify expression of antigen-presentation as a positive prognostic factor, but this is a well-established trend across many different tumor types, and has been noted specifically for TNBC in multiple previous publications (for example, see PMIDs 28817603, 26980599, 30193111). Forero et al. is actually referenced in the discussion, but it was not noted that this previous study was similar in size to the current one and included bulk RNA seq with confirmatory laser capture microdissection and single plex IHC, the latter being something that is notably missing here. The compartment-specific nature of this effect is interesting, but given the limitations of the DSP antibody platform, the authors are not able to follow-up further on this trend. The correlation analyses are likewise not very informative. Overall, this is a well-designed study with rigorous statistical analyses, but the novelty of the findings is limited.

Major Concerns:

1. One of the major findings is that proteins associated with antigen presentation are enriched in the intraepithelial segments. While HLA-DR can be expressed in both tumor cells or APCs, the authors are not able to identify the cell population responsible for this increased expression. The authors cite one of the prior studies demonstrating that intraepithelial HLA-DR was associated with better clinical outcome, so it is unclear what additional insight this study presents. Additionally, multiple avenues exist to probe this directly (i.e. IF with HLA-DR, CD45, and PanCK) that would not require the expensive and cumbersome techniques employed here.

RESPONSE: We agree with the reviewer's points about HLA-DR. However, as the reviewer mentioned, prior studies evaluating the relationship with HLA-DR and clinical outcomes were performed with either a very small sample size, or used single or low-plex techniques for their protein approaches. In our revised manuscript, we have leveraged the high-plex nature of the GeoMX platform to interrogate the spatial relationship and co-localization of multiprotein immune-based features (antigen-presenting including HLA-DR and T cell activation) as well as individual immune checkpoint proteins (PD-L1 and IDO-1) with outcomes in these 2 TNBC cohorts. We have included a number of findings (summarized in point below) about these targets, providing insights about intrinsic spatial immunobiology of TNBC and informative for rational biomarker strategies for pembrolizumab or other PD-(L)1-based targeted therapeutics. Nevertheless, ultimately, we agree with the reviewer that a limitation of the GeoMX platform is the lack of single cell resolution. We are actively working with Nanostring on its novel spatial single cell platform (CosMX) to address this critical technological gap.

2. While the authors perform sound statistical analysis, the motivation behind performing these analyses is unclear. For example, the correlation analysis shown in Table 3 largely recapitulates known biology (i.e. it is not informative that T cell and APC markers are correlated)

RESPONSE: We agree with the reviewer and these analyses have been removed from the revised manuscript. Beyond new sTIL data and multivariate analyses in both cohorts, data from new analyses have been added in which we make use of the high-plex simultaneous quantitation of numerous protein targets with DSP to evaluate the relationship between functional immune modules such as antigen-presenting activity (the APC eigengene score), T-cell activation status (the TCA score), along with clinically-actionable immune checkpoint proteins such as PD-L1, and IDO1 in individual tumors.

In both cohorts, these data indicate that improved clinical outcomes are closely tied to both spatial context and co-expression of immune features: we determined that several actionable immune-features are prognostic only in the intraepithelial compartments and only when co-localized. As one might expect, tumors with both high intraepithelial APC and TCA scores had better clinical outcomes than those with low scores. However, we did identify subsets of TNBC with high intraepithelial APC or TCA scores, but not both, and neither high APC nor high intraepithelial TCA scores alone were associated

with improved outcomes. Moreover, we have continued to characterize the prognostic components of the antigen-presenting feature as detailed below.

For PD-L1 and IDO-1, tumors with high intraepithelial PD-L1 levels required high APC scores to be associated with significantly better RFS, whereas high intraepithelial IDO-1 were associated with better outcomes irrespective of APC score. We argue that these observations are important for several reasons: in the early-stage setting of TNBC, pembrolizumab has been approved as the first-line neoadjuvant therapy irrespective of tumoral PD-L1 status, underscoring the urgent need for clinically-relevant biomarkers of the PD-(L)1 axis to improve patient selection. In contrast, in the setting of advanced disease, pembrolizumab is FDA-approved only for patients with TNBC with high-expression of PD-L1 (PharmDX 22C3 assay with CPS ≥ 10), and shows only modest therapeutic benefit. PD-L1 protein can be expressed on several immune cell types (macrophages, dendritic cells and lymphocytes) and less frequently, tumor cells in TNBC, and the scoring system for the PharmDX 22C3 immunohistochemical assay does not discriminate among PD-L1-expressing immune cell types. **Thus, there is a critical knowledge gap of which PD-(L)1-expressing cell population(s) confer or predict responsiveness to pembrolizumab in early-stage or mTNBC.** Our observations about PD-L1 and IDO-1 with APC immune features identified spatial compartment and co-localization requirements for these targets to have prognostic value. We argue this is a first but key step toward improving biomarker-based patient selection for targeted PD-(L)1 therapies, or paradoxically, therapeutic de-escalation in select patient subsets with early-stage TNBC, as both intraepithelial PD-L1 and IDO-1 enrichment were associated with better clinical outcomes in these cohorts.

3. The analysis in the “Spatial context determines immune protein association with outcome” section is quite repetitive. The authors first identify the proteins which are associated with RFS (Table 1). The authors then plot the expression of these proteins, and find differences in expression between patients with good outcome and patients with poor outcome (Figure 1). However, these proteins were selected because they were associated with outcome, so the purpose of this figure is unclear.

RESPONSE: The purpose of Figure 1 is to show that the distribution of expression of some of these key proteins in the intraepithelial segments is different from the distribution in the stroma. This is an important point: the expression profiles in the intraepithelial segments are not simply a reflection of the profiles in the adjacent stroma.

Next, the authors take these same set of proteins, which we know are associated with survival, and demonstrate that there are significant ORs when binning patients based off their expression level (Table 2). Finally, using these same proteins, the authors construct Kaplan-Meier curves, and find differences in outcome (Figure 2), which is once again not surprising

RESPONSE: We agree with the reviewer. Much of these repetitive data have been removed from the revised manuscript. However, we submit that there is some value in giving ORs, HRs, and KM curves for

the same data. ORs reflect the risk of recurrence within a defined time frame, 5 years in this case. HRs reflect the risk of recurrence when time to event is considered as a continuous variable, and KM curves give a good graphical representation of percent of patients who recur as a function of time. We have used all of these analyses for the data shown in Figure 3. However, most of these data are now in Supplementary data.

Minor Concerns:

1. The LFC values are fairly modest, even for markers which are explicitly selected for ROI creation. Is there a reason that a greater difference in expression was not observed?

RESPONSE: We can only speculate concerning this issue. A likely explanation would be cellular heterogeneity. It is plausible that the observed fold changes are averaged within a population of cells, of which only a subset are directly involved in the response. Single cell spatial analysis will be required to resolve this issue, and we are currently working with NanoString to develop this technology.

2. The authors use fold change throughout the paper, which can be unstable for low-expressing proteins. A high fold change could be driven by low expression, which may lead to misleading conclusions. In particular, the authors state that PD-L2 was the only protein associated with durable recurrence free survival in any stromal segment class, and also state that PD-L2 was expressed at low levels. Along these lines, we have found PD-L2 to in the vast majority of cases have weak and non-specific staining with all commercial clones we've tested, making this finding further suspect without confirmatory single-plex imaging. This significant fold change could be due to low expression. It would be helpful for the authors to include the baseline expression levels as well, or a violin plot.

RESPONSE: Actually, we use fold change only in Table 1 and the supplementary figures that describe the difference between the different segment categories. We defined background as the mean plus two standard deviations of the abundance of two negative controls, and we excluded from analysis any protein for which the mean abundance was less than background. Only 29 out of 45 proteins met this criterion for FinXX. Regarding PD-L2, it is a difficult protein to analyze, giving a relatively low signal, as shown in Supplementary Table 4. Whether this is due to low abundance (likely) or a low affinity antibody cannot be determined for our data. Consequently, we have reported the PD-L2 data, which may be quite interesting; but we have not pursued PD-L2 further than this basic description.

3. It would be helpful to include significance values in Figure 1.

RESPONSE: I am unsure how to define the null hypothesis for histogram data such as that shown in Figure 1.

4. The authors state that the limited correlation between total CD45 levels and the other immune proteins they identified means that there is independent prognostic information in these proteins

beyond sTIL levels (top of page 16). However, this claim requires a multivariate model showing independent significance to verify that this is the case

RESPONSE: As requested by the reviewers, we have deleted the CD45 analysis and replaced this aspect of the study with clinical TILs as a co-variable in multivariable analysis.

5. It is surprising that CD20 does not correlate with CD45, since presumably cells that are expressing CD20 (B cells) are also expressing CD45. Some more explanation about these findings would be helpful. These results could potentially be biased by non-specific antibody binding.

RESPONSE: The correlative studies have been removed from the revised text. We have not focused on CD20 because our analyses did not seem to have consistent outcome in FinXX and the TMA samples. The reasons for this are unclear.

6. Similarly, B cells are antigen presenting cells and express HLA-DR, and it is a little surprising CD20 does not correlate with HLA-DR. It seems in Figure 3a that they are more coupled in the stroma. It would be helpful to comment on these findings in more detail.

RESPONSE: I would not necessarily expect that a marker for any single cell type (e.g. CD20) would correlate with bulk HLA-DR expression, which is derived from many different cellular sources. In any event, as suggested by the reviewers, we have deleted these correlative expression analyses from the revised manuscript.

7. In the differential expression plots (Supplementary Figure 2 and 5), it would be helpful to sort by average fold change to be able to see patterns more clearly.

RESPONSE: We elected to sort the data alphabetically, by protein name, to maintain consistency from one supplementary table to the next. However, these are excel spreadsheets and can be sorted by the reader in any manner of his or her choosing.

8. The authors state that they observed similar LFCs in the Mayo and FinXX cohort, but this is difficult to evaluate since they are presented in different supplementary figures. A single figure which has one cohort on the X axis and the other cohort on the Y axis would be an easier way to visually assess the similarities between the two datasets

RESPONSE: We agree with the reviewer. We have now included log fold changes for the key proteins of interest in FinXX and the TNBC TMA cohort in supplementary table 6.

9. While the authors mention in the discussion that the GeoMx platform cannot provide single-cell level data and therefore cannot be used to assess single-cell level expression, it is possible to use the

fluorescent images used to guide region selection to get an approximation for the number of CD45+ and cytokeratin+ cells (i.e. immune and tumor cells). These counts could be informative to include to assess whether there is more immune infiltrate, or a similar number of cells with cells expressing a higher level of protein.

RESPONSE: This is a vitally interesting question, but one than cannot directly be answered using GeoMx technology. We are very actively pursuing this question using emerging CosMx technology, which has single cell spatial resolution to enable direct interrogation of the cell types of interest.

10. The authors have not made their code or data available. This should be remedied prior to publication.

RESPONSE: We are in the process of uploading the NanoString DSP data to GEO. No significant new code was written for these analyses, most of which were run with open source R code.

REVIEWERS' COMMENTS

Reviewer #1 (Remarks to the Author):

The concerns of this reviewer have been addressed.

Reviewer #3 (Remarks to the Author):

The authors have made significant modifications to the manuscript and they have addressed most of the key issues that I raised. We still differ on the point of nomenclature, but I am OK with the revised version

Reviewer #4 (Remarks to the Author):

We would like to start off by citing our frustration that the authors failed to address many of the points that we raised in the initial review. We feel just as strongly as before that these points need to be addressed.

1. In response to our comment that it would be helpful to include significance values in Figure 1 (Minor Concern #3), the authors stated they were unsure of how to do so. There are many statistical tests designed to compare two distributions, for example, the Kolmogorov-Smirnov test. There are various online resources for choosing the correct test and applying such tests in R or Python. This is a very reasonable request and we feel it is unacceptable to not provide this information.

2. We suggested that visualization and interpretation of the differential expression plots would be helped by sorting by average fold change, which is the conventional way this data are presented (Minor Concern #7). The authors responded that they “elected to sort the data alphabetically, by protein name, to maintain consistency from one supplementary table to the next.” It is fine to sort the supplementary tables alphabetically, but the purpose of figures is for the reader to easily interpret and extract the relevant information to understand the text. We feel that sorting by average fold change would help do so, so the reader does not have to hunt in the figure for the proteins mentioned in the text. The authors further state that the tables “are excel spreadsheets and can be sorted by the reader in any manner of his or her choosing.” It is good that these spreadsheets are provided, but it is not the responsibility of the reviewer or the reader to extract the relevant information from information-dense tables to understand the text.

3. Similarly, regarding low expression impacting fold change, we suggested that the authors include a figure showing baseline expression (Minor Concern #2). The authors indicated a Supplementary Table. The table has a lot of information, and it is not the reader/reviewer’s responsibility to extract the relevant information from the large amount of information in the supplementary tables. As the authors still highlight PD-L2 in the text, we believe the baseline expression of PD-L2 (and other proteins that are highlighted in the text) should be displayed in an easily interpretable figure.

4. We suggested to the authors to use the fluorescent images to get an approximation for the number of CD45+ and cytokeratin+ cells (Minor Concern #9). The authors responded that this “cannot directly be answered using GeoMx technology.” This response is confusing to us. As shown in a supplemental figure, the authors have access to the fluorescent images used to generate the masks. Therefore, it should be possible to use these fluorescent images for additional analysis, as we suggested. Since the GeoMx platform is based on IF microscopy, this is nonsensical response.

5. In response to our comment that the code should be made publicly available (Minor Concern #10), the authors stated that “No significant new code was written for these analyses, most of which were run with open source R code.” The vast majority of the findings and figures in this manuscript are generated using computational analysis, and therefore we feel it is necessary to make such code

transparent. It is common practice (and often even required) for code to be deposited, even if the analyses were run with open source R code. We also note it is required by all Nature journals that any code be available at the time of review. Not providing the code is not acceptable.

The main change that Carter et al. made was the addition of analysis showing that intraepithelial PD-L1 was only a prognostic marker when there were high levels of antigen-presentation or T cell activation, while intraepithelial IDO-1 was a prognostic marker agnostic of antigen-presentation. Furthermore, the authors show that HLA-DR and CD40 were associated with improved survival in only the epithelial compartment, while IDO-1 was associated with improved survival in the intraepithelial or stromal compartment. In our original review, we found that the novelty of the findings was limited. Even with the added analysis, we would argue that this assessment stands. In general, the authors have shown that proteins that were previously known to correlate with improved survival indeed correlate with improved survival. Other concerns are listed below:

1. Multiple immune proteins are associated with improved overall survival when comparing upper tertile to bottom two tertiles (Table 2A). It would be helpful to have better contextualization of what markers are known to be prognostic based on previous literature. The main takeaway from this table is that immune infiltration with inflammatory subsets is associated with improved survival, which is well established.

2. The utility of the eigengene score is not clear. The survival ratios appear to be similar for the individual proteins which make up the eigengene calculation. For example, the hazard ratios for CD11c, HLA-DR, CD40 are all significant in the intraepithelial compartment as shown in Table 2A. Therefore, it is not clear what the benefit of aggregating multiple proteins into an eigengene score is. Furthermore, "eigengene" should be renamed. While eigengenes are commonly used in the literature, since this study is not looking at genes, the terminology is not appropriate here.

We also had issues accessing and interpreting files provided by the authors:

1. We could not open the PowerPoint file provided ("313942_1_supp_7023255_rkbdqf") that presumably contained the supplementary figures. The error message was "PowerPoint can't open this file because its file extension has changed. Make sure the format of the file matches the file extension." We tried using both Mac and PC. Therefore, we could not evaluate the supplementary figures.
2. The supplementary tables are not clearly labeled (e.g. "313942_1_data_set_7023256_rkbdqf", instead of "Supplementary Table x"). As such, the responsibility fell on the reviewer to figure out which tables the text was referring to, making interpretation difficult.

Reviewers' response to revised manuscript

Reviewer's Comments:

Reviewer #1 (Remarks to the Author)

The concerns of this reviewer have been addressed.

Reviewer #3 (Remarks to the Author)

The authors have made significant modifications to the manuscript and they have addressed most of the key issues that I raised. We still differ on the point of nomenclature, but I am OK with the revised version

Reviewer #4 (Remarks to the Author)

We would like to start off by citing our frustration that the authors failed to address many of the points that we raised in the initial review. We feel just as strongly as before that these points need to be addressed.

Response: I note initially that point 1 (below) refers to Minor Concern #3, point 2 refers to Minor Concern #7, point 3 to Minor Concern #2, point 4 to Minor Concern #9, and point 5 to Minor Concern #10. Many of these minor concerns relate to the manner in which our data are presented in supplementary material, which admittedly contains a huge amount of data, from which the figures and tables in the text were derived.

1. In response to our comment that it would be helpful to include significance values in Figure 1 (Minor Concern #3), the authors stated they were unsure of how to do so. There are many statistical tests designed to compare two distributions, for example, the Kolmogorov-Smirnov test. There are various online resources for choosing the correct test and applying such tests in R or Python. This is a very reasonable request and we feel it is unacceptable to not provide this information.

Response: We have run KS analysis on the curves shown in Figure 1. The results, as expected, are in all respects consistent with the data in Table 1. Note that among these features, only IDO1 was significant in both tumor (i.e., intraepithelial) and stroma segments. We have modified figure 1 to include the KS p-values.

Terms	KS test P value Finxx tumor	KS test P value Finxx stroma
B2M	0.049	0.109
CD11c	0.020	0.632
CD20	0.215	0.872
CD40	0.002	0.109
CD56	0.002	0.218
GZMB	0.218	0.632
HLA_DR	0.020	0.872
ICOS	0.020	0.109
IDO1	0.007	0.007
PD_L2	0.621	0.632

2. We suggested that visualization and interpretation of the differential expression plots would be helped by sorting by average fold change, which is the conventional way this data are presented (Minor Concern #7). The authors responded that they “elected to sort the data alphabetically, by protein name, to maintain consistency from one supplementary table to the next.” It is fine to sort the supplementary tables alphabetically, but the purpose of figures is for the reader to easily interpret and extract the relevant information to understand the text. We feel that sorting by average fold change would help do so, so the reader does not have to hunt in the figure for the proteins mentioned in the text. The authors further state that the tables “are excel spreadsheets and can be sorted by the reader in any manner of his or her choosing.” It is good that these spreadsheets are provided, but it is not the responsibility of the reviewer or the reader to extract the relevant information from information-dense tables to understand the text.

Response: This point refers, I believe, to the supplementary data figures that compare expression of the key proteins in various segments (e.g., intraepithelial vs stroma as a function of CD45 abundance in the adjacent stroma). I do not wish to appear to be argumentative, I am not aware of a convention that holds that data should be presented in rank order of fold-change. Nor do I understand why plotting the data rank ordered as a function of fold-change is more informative than plotting the data as a function of protein name in alphabetical order. However, we have provided an additional set of supplementary tables that force ranks each of the features for each of the figures, rank ordered in terms of fold-change. We note that such data are already included in supplementary tables, but we have extracted the fold change, 95% Cis, and p-values to assist the reader in evaluating our data.

3. Similarly, regarding low expression impacting fold change, we suggested that the authors include a figure showing baseline expression (Minor Concern #2). The authors indicated a Supplementary Table. The table has a lot of information, and it is not the reader/reviewer’s responsibility to extract the relevant information from the large amount of information in the supplementary tables. As the authors still highlight PD-L2 in the text, we believe the baseline expression of PD-L2 (and other proteins that are highlighted in the text) should be displayed in an easily interpretable figure.

Response: This concern puzzles me, since we did give baseline expression data in supplementary table 3, which does not appear to be to be a particularly complicated table to interpret. One may question the reader's ability to make much out of a box plot with 59 boxes for each of the two segments analyzed. For my part, I would prefer a table that I can search; but the reviewer feels otherwise. I appreciate the reviewer's concern that there are a lot of data in some of these tables, but the amount of data generated is inherent in the nature of the experimental design. We have endeavored to provide the readers/reviewers of this manuscript with all the information that they might need to replicate our findings, should they choose to re-run our analyses or to analyze the data using some other statistical approach. I am at a loss as to how to simplify these tables without omitting the data that we used to run our analyses. In short, what we have attempted to do is to provide figures and tables in the text which summarize our key findings, while providing the complete data upon which our interpretations are based in supplementary tables. All that being said, we have now included new supplementary data that include the fold change data from the individual supplementary figures (see point 2). I submit that these new table, although redundant to the data in the figures and tables, will relieve the reviewer of the effort of having to sort through the data in our supplementary table to obtain a numerical, rather than graphical, representation of our results.

4. We suggested to the authors to use the fluorescent images to get an approximation for the number of CD45+ and cytokeratin+ cells (Minor Concern #9). The authors responded that this “cannot directly be answered using GeoMx technology.” This response is confusing to us. As shown in a supplemental figure, the authors have access to the fluorescent images used to generate the masks. Therefore, it should be possible to use these fluorescent images for additional analysis, as we suggested. Since the GeoMx platform is based on IF microscopy, this is nonsensical response.

Response: I must point out that in the original version of this manuscript we included analysis of CD45+ cells from quantitative image analysis of our samples. We relied on such image analysis data as a surrogate marker of stromal tumor-infiltrating lymphocytes (sTILs), and we used these data in multivariable analysis to determine which of the several features of interest were significant prognostic indicators when CD45+ cells were included as a covariable. The reviewers objected to the use of CD45+ cells as surrogates for sTILs and insisted that we must obtain and incorporate data from conventional histopathological analysis of sTILs instead of CD45+ cells as a surrogate. This we did. We can, if the editor thinks it appropriate to do so, go back and re-introduce the image analysis data for CD45+ cells. We have not attempted to quantify cytokeratin-positive cells, and I am unsure of what would be learned by measuring the number of cytokeratin-positive cells in the cytokeratin-positive (intraepithelial) niche. It could be done, but it is not trivial to do so.

5. In response to our comment that the code should be made publicly available (Minor Concern #10), the authors stated that “No significant new code was written for these analyses, most of which were run with open source R code.” The vast majority of the findings and figures in this manuscript are generated

using computational analysis, and therefore we feel it is necessary to make such code transparent. It is common practice (and often even required) for code to be deposited, even if the analyses were run with open source R code. We also note it is required by all Nature journals that any code be available at the time of review. Not providing the code is not acceptable.

Response: We have downloaded all of the code that was used to Github, a publically available source for such things. A Github link has been provided in Methods.

The main change that Carter et al. made was the addition of analysis showing that intraepithelial PD-L1 was only a prognostic marker when there were high levels of antigen-presentation or T cell activation, while intraepithelial IDO-1 was a prognostic marker agnostic of antigen-presentation. Furthermore, the authors show that HLA-DR and CD40 were associated with improved survival in only the epithelial compartment, while IDO-1 was associated with improved survival in the intraepithelial or stromal compartment. In our original review, we found that the novelty of the findings was limited. Even with the added analysis, we would argue that this assessment stands. In general, the authors have shown that proteins that were previously known to correlate with improved survival indeed correlate with improved survival. Other concerns are listed below:

1. Multiple immune proteins are associated with improved overall survival when comparing upper tertile to bottom two tertiles (Table 2A). It would be helpful to have better contextualization of what markers are known to be prognostic based on previous literature. The main takeaway from this table is that immune infiltration with inflammatory subsets is associated with improved survival, which is well established.

Response: We have performed another literature search to determine which of these protein features (CD40, HLA-DR and CD11c) has been previously independently associated with prognosis in TNBC and the discussion is updated to reflect this change.

2. The utility of the eigengene score is not clear. The survival ratios appear to be similar for the individual proteins which make up the eigengene calculation. For example, the hazard ratios for CD11c, HLA-DR, CD40 are all significant in the intraepithelial compartment as shown in Table 2A. Therefore, it is not clear what the benefit of aggregating multiple proteins into an eigengene score is. Furthermore, "eigengene" should be renamed. While eigengenes are commonly used in the literature, since this study is not looking at genes, the terminology is not appropriate here.

Response: We changed the name to "eigenprotein". We submit that the virtue of such aggregated features lies in their ability to provide a surrogate assessment of the activity of a biological process. In a sense, the eigenprotein score is similar to a pathway activation score in more conventional bioinformatic analysis. Such scores are widely accepted as representative of biological process activity.